# POCO: Scalable Neural Forecasting through Population Conditioning

**Yu Duan**[1,5]  **Hamza Tahir Chaudhry**[2]  **Misha B. Ahrens** [3]  **Christopher D. Harvey**[4]
**Matthew G. Perich** [6,7]    **Karl Deisseroth**[8*]    **Kanaka Rajan**[4,5*]
[1]EECS, MIT    [2]SEAS, Harvard University    [3]Janelia Research Campus, HHMI
[4] Harvard Medical School    [5]Kempner Institute    [6]Université de Montréal
[7] Mila - Quebec AI Institute    [8] Stanford University    [*]Corresponding Authors
deissero@stanford.edu, kanaka_rajan@hms.harvard.edu

## Abstract

Predicting future neural activity is a core challenge in modeling brain dynamics, with applications ranging from scientific investigation to closed-loop neurotechnology. While recent models of population activity emphasize interpretability and behavioral decoding, neural forecasting—particularly across multi-session, spontaneous calcium recordings—remains underexplored. We introduce POCO, a unified forecasting model that combines a lightweight univariate forecaster with a population-level encoder to capture both neuron-specific and brain-wide dynamics in calcium imaging recordings. Trained across five calcium imaging datasets spanning zebrafish, mice, and *C. elegans*, POCO achieves state-of-the-art accuracy at cellular resolution in spontaneous behaviors. After pre-training, POCO rapidly adapts to new recordings with minimal fine-tuning. Notably, POCO's learned unit embeddings recover biologically meaningful structure—such as brain region clustering—without any anatomical labels. Our comprehensive analysis reveals several key factors influencing performance, including context length, session diversity, and preprocessing. Together, these results position POCO as a scalable and adaptable approach for cross-session neural forecasting and offer actionable insights for future model design. By enabling accurate, generalizable forecasting models of neural dynamics across individuals and species, POCO lays the groundwork for adaptive neurotechnologies and large-scale efforts for neural foundation models. Code is available at `https://github.com/yuvenduan/POCO`.

## 1  Introduction

The ability to predict future states from the past is a critical benchmark for models of complex systems such as the brain [32]. Models capable of rapidly and accurately forecasting future neural activity across large spatial-temporal scales—rather than merely fitting historical data—are critical for applied technologies such as closed-loop optogenetic control [19]. Here, we focus on a time-series forecasting (TSF) setup: Given a recent history of measured neural activity, can we predict the neural population dynamics in the near future?

Recently, the increasing ability to record from large populations of neurons has inspired a wide range of models for neural population dynamics [15, 27, 46, 60]. However, much of this work has focused on interpreting latent features or decoding behavior, while forward prediction of neural activity remains relatively underexplored. While some recent work has begun to explore forward modeling of spiking data [17, 2, 56, 38], these approaches are not designed for calcium imaging and have not been systematically compared with state-of-the-art models from time-series forecasting. Moreover, existing studies also tend to focus on short, trial-based data from individual animals in controlled tasks, but a comprehensive understanding of neural dynamics benefits from whole-brain recordings

39th Conference on Neural Information Processing Systems (NeurIPS 2025).

during spontaneous, task-free behaviors. Furthermore, growing evidence for shared neural motifs across individuals [42, 12, 37], along with the rise of large-scale, multi-animal datasets, motivates the development of *foundation models*—models trained across individuals that generalize to unseen subjects [5, 6, 2]. However, classical models of population dynamics predominantly focus on fitting single-session recordings [40, 27, 15], limiting their ability to utilize larger datasets and capture common motifs shared across animals.

To address these gaps, we developed POCO (POpulation-COnditioned forecaster), a unified predictive model for forecasting spontaneous, brain-wide neural activity. Trained on multi-animal calcium imaging datasets, POCO predicts cellular-resolution dynamics up to ∼15 seconds into the future. It combines a simple univariate forecaster for individual neuron dynamics with a population encoder that models the influence of global brain state on each neuron, using Feature-wise Linear Modulation (FiLM) [39] to condition forecasts on population-level structure. For the population encoder, we adapt POYO [5]—originally developed for behavioral decoding in primates—to summarize high-dimensional population activity across sessions. We benchmark POCO against standard baselines and state-of-the-art TSF models trained on five datasets from zebrafish, mice, and *C. elegans*.

In sum, our key contributions are: (1) We introduce POCO, a novel architecture that combines a local forecaster with a population encoder, for cellular-level neural dynamics forecasting. (2) We benchmark the forecasting performance of a wide range of models on five diverse calcium imaging datasets spanning different species, with a focus on neural recording during spontaneous behaviors. (3) We demonstrate that POCO scales effectively with longer recordings and additional sessions, and pre-trained POCO can quickly adapt to new sessions. (4) We conduct extensive analyses on factors that affect performance, including context length, dataset pre-processing, multi-dataset training, and similarity between individuals. These analyses provide critical insights for future work on multi-session neural forecasting.

## 2 Related Work

**Neural Foundation Models.** A growing body of work aims to develop unified models trained on neural data across multiple subjects, tasks, and datasets. This foundation model approach has been applied to spiking data in primates [5, 53, 52, 56, 2], human EEG and fMRI recordings [8, 10, 48], and calcium imaging in mice [6]. While much of this work emphasizes improving behavioral decoding performance [5, 6, 53], some studies have explored forward prediction [17, 2, 56, 38]. However, these efforts are largely confined to spiking data recorded during short, trial-based motor or decision-making tasks. Neural prediction has also been explored in *C. elegans* [44], but the setup is limited to next-step prediction using autoregressive models.

**Models of Population Dynamics.** To understand high-dimensional population dynamics, one line of work has focused on inferring low-dimensional latent representations from observations with models including RNNs [15, 35, 40], switching linear dynamical systems [18, 27], sequential variational autoencoder (VAE) [46, 57, 37, 60], and latent diffusion models [22]. While some of these models could, in theory, be adapted for forecasting, their focus to date has been to gain interpretable insights, especially in constrained neuroscience tasks.

**Time-Series Forecasting.** Time series forecasting is a general problem that emerges in a variety of domains [51, 29, 58, 32]. Deep learning has led to a wide range of TSF architectures, including RNNs [43], temporal convolutional networks (TCNs) [28, 33], and Transformer-based models [58, 59, 47]. Simpler models—such as MLP-based architectures [9, 54, 13] and even linear models [49, 55]—often perform competitively or even outperform more complex alternatives. In neuroscience, Zapbench [32] is a recent benchmark for forecasting, though recordings are limited to a single larval zebrafish.

## 3 Method

### 3.1 Problem Setup

In this work, we consider a multi-session TSF problem. For a session $j \in [S]$, we use $\mathbf{x}_t^{(j)} \in \mathbb{R}^{N_j}$ to denote the neural activity at time step $t$, where $N_j$ is the number of neurons recorded in session $j$. Given population activity of the last $C$ time steps $\mathbf{x}_{t-C:t}^{(j)} := \mathbf{x}_{t-C,\dots,t-1}^{(j)} \in \mathbb{R}^{C \times N_j}$, we hope to find

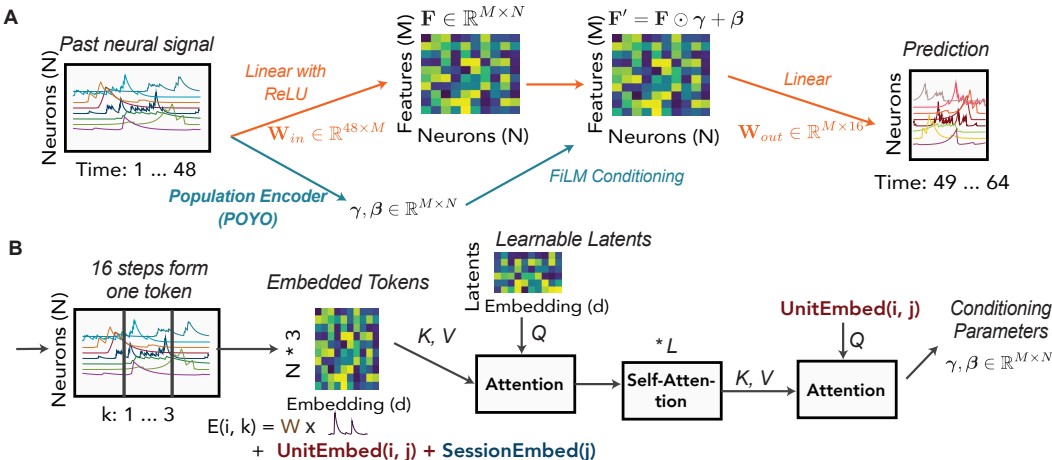

Figure 1: *Model Architecture.* (A) POCO combines a univariate MLP forecaster (orange part) and a population encoder that conditions the MLP (blue part). This is a schematic for illustration only; traces and feature maps shown are not actual model input, output, or embedding. (B) The population encoder is adapted from POYO [5]. We split the trace of each neuron into several tokens, encode the tokens with POYO, and then use unit embedding to query the conditioning parameters. See the Method section for more details.

a predictor $f$ that forecasts the next $P$ steps and minimizes the mean squared error $L$:

$$f\left(\mathbf{x}_{t-C:t}^{(j)}, j\right) = \tilde{\mathbf{x}}_{t:t+P}^{(j)}, \quad L(f) = E_{j,t}\left[\frac{1}{PN_i}\|\tilde{\mathbf{x}}_{t:t+P}^{(j)} - \mathbf{x}_{t:t+P}^{(j)}\|_F^2\right], \tag{1}$$

where we use $\|\cdot\|_F$ denotes Frobenius norm. In most experiments, we use $C = 48$ and $P = 16$. Importantly, the number of neurons $N_j$ varies across sessions, and neurons in different animals do not have one-to-one correspondences; yet, the forecasting problems for different sessions are closely linked due to the similarity in neural dynamics between animals [5, 6, 2], which distinguishes this setting from standard multivariate TSF setups.

## 3.2 Population-Conditioned Forecaster

We first introduce the overall framework of POCO (Figure 1A). Consider an MLP forecaster with hidden size $M = 1024$ that takes past population activity $\mathbf{x}_{t-C:t}^{(j)} \in \mathbb{R}^{C \times N_j}$ as input:

$$f_{\text{MLP}}\left(\mathbf{x}_{t-C:t}^{(j)}\right) = \mathbf{W}_{out} \text{ReLU}(\mathbf{W}_{in}\mathbf{x}_{t-C:t}^{(j)} + \mathbf{b}_{in}) + \mathbf{b}_{out}, \tag{2}$$

where $\mathbf{W}_{in} \in \mathbb{R}^{M \times C}$ and $\mathbf{W}_{out} \in \mathbb{R}^{P \times M}$ are weight matrices. The MLP forecaster is *univariate*, meaning that the prediction for a neuron only depends on its own history, capturing individual auto-correlative properties and simple temporal patterns. Prior work has shown that these simple univariate forecasters perform surprisingly well even for multivariate data [47, 13, 54, 55].

Building on the MLP forecaster, we add a population encoder $g$ that modulates the prediction of the MLP through Feature-wise Linear Modulation (FiLM) [39]. Specifically, the population encoder gives the conditioning parameters $g(\mathbf{x}_{t-C:t}^{(j)}, j) = (\gamma, \beta)$, where $\gamma, \beta \in \mathbb{R}^{M \times N_j}$ are of the same shape as the hidden activations in MLP. We then define POCO as

$$f_{\text{POCO}}\left(\mathbf{x}_{t-C:t}^{(j)}\right) = \mathbf{W}_{out}\left[\text{ReLU}(\mathbf{W}_{in}\mathbf{x}_{t-C:t}^{(j)} + \mathbf{b}_{in})\odot\gamma + \beta\right] + \mathbf{b}_{out}, \tag{3}$$

where $\odot$ denotes element-wise multiplication. Intuitively, the FiLM conditioning allows the population encoder to modulate how each neuron's past activity is interpreted, effectively tailoring the MLP forecaster to the broader brain state at each time point. This enables the model to account for context-dependent dynamics while maintaining neuron-specific predictions.

## 3.3 Population Encoder

We then need to define a population encoder $g$ capable of modeling how the population state influences each neuron. To this end, we adapt a recent architecture, POYO [5, 6], which combines the Perceiver-

IO architecture [21] and a tokenization scheme for neural data (Figure 1B). Specifically, for each neuron $i$, we partition the trace into segments of length $T_C = 16$ and each segment forms a token, creating $C/T_C = 3$ tokens for each neuron. Then for each token $k \in [C/T_C]$ of each neuron $i \in [N_j]$, we define the embedding $E(i, k) \in \mathbb{R}^d$ as

$$E(i, k) = \mathbf{W} x_{r_k - T_C : r_k, i}^{(j)} + \mathbf{b} + \text{UnitEmbed}(i, j) + \text{SessionEmbed}(j), \qquad (4)$$

where $\mathbf{W} \in \mathbb{R}^{d \times T_C}$ is a linear projection, $r_k = t - C + kT_C$ defines the temporal boundary of a token, and both $\text{UnitEmbed}(i, j)$ and $\text{SessionEmbed}(j)$ are learnable embeddings in $\mathbb{R}^d$. Intuitively, after learning, the unit embedding can define the dynamical or functional properties of the neuron, while the session embedding can account for different recording conditions (e.g., sampling rates, raw fluorescence magnitude) in different sessions. These token embeddings are arranged into a matrix $\mathbf{E} \in \mathbb{R}^{CN_j/T_C \times d}$, which is then processed by Perceiver-IO [21]. Specifically, we use $N_L = 8$ learnable latents $\mathbf{L}_0 \in \mathbb{R}^{N_L \times d}$ as the query for the first layer. $\mathbf{L}_0$ is shared across sessions. After $L$ self-attention layers, the final attention layer uses the unit embedding $\mathbf{U}_j = \text{UnitEmbed}(*, j) \in \mathbb{R}^{N_j \times d}$ as queries to extract conditioning parameters $(\boldsymbol{\gamma}, \boldsymbol{\beta})$:

$$
\begin{aligned}
\mathbf{L}_1 &= \text{Attention}_0(Q = \mathbf{L}_0; K, V = \mathbf{E}) \in \mathbb{R}^{N_L \times d}, \\
\mathbf{L}_{l+1} &= \text{Attention}_l(Q, K, V = \mathbf{L}_l) \in \mathbb{R}^{N_L \times d}, l \in [L], \\
\mathbf{L}_{L+2} &= \text{Attention}_{L+1}(Q = \mathbf{U}_j; K, V = \mathbf{L}_{L+1}) \in \mathbb{R}^{N_j \times d}, \\
\boldsymbol{\beta} &= \mathbf{W}_\beta \mathbf{L}_{L+2}^T + \mathbf{b}_\beta, \quad \boldsymbol{\gamma} = \mathbf{W}_\gamma \mathbf{L}_{L+2}^T + \mathbf{b}_\gamma,
\end{aligned}
\qquad (5)
$$

where $\text{Attention}(Q, K, V)$ is a multi-head attention layer [50], $\mathbf{W}_\gamma, \mathbf{W}_\beta \in \mathbb{R}^{M \times d}$ are learned weight matrices. We have $L + 2$ attention layers in total, where $L = 1$ in most experiments. Following POYO, we use rotary position embedding [45] (details are omitted above for simplicity). One advantage of the Perceiver-IO architecture is that the time complexity only scales linearly with the number of neurons, allowing the model to efficiently scale to recordings of large neural populations. Although we refer to individual neurons throughout, the same framework can also be applied to reduced representations of neural activity, such as principal components (PCs).

Our population encoder has two notable differences from the recent POYO+ model, which is also designed for calcium data [6]. First, POYO+ is designed for behavioral decoding and thus the unit embedding is only used for tokenization. In contrast, here, unit embedding is reused in the last layer to query how the population drives each neuron. Second, in POYO+, $T_C$ is always fixed to 1, which creates a massive number of tokens when the context length $C$ and the number of neurons $N_j$ are large. We discuss the effect of $T_C$ in Figure S15.

## 4 Benchmark

### 4.1 Datasets

Table 1: *Overview of the five datasets.* $f_s$ is the sampling frequency. The number of neurons, recording length, and sampling frequency vary by session, the approximate average is shown here.

| Species | Lab | #Sessions | #Neurons | #Steps | $f_s$ | Ca$^{2+}$ Indicator |
|---|---|---|---|---|---|---|
| Larval Zebrafish | Deisseroth[1] | 19 | 11K | 4.3K | 1.1Hz | GCaMP6s |
| Larval Zebrafish | Ahrens[11] | 15 | 77K | 3.9K | 2.1Hz | GCaMP6f |
| Mice | Harvey[40, 3] | 12 | 1.6K | 15K | 5.4Hz | GCaMP6s |
| *C. elegans* | Zimmer[23] | 5 | 126 | 3.1K | 2.8Hz | GCaMP5K |
| *C. elegans* | Flavell[4] | 40 | 136 | 1.6K | 1.7Hz | GCaMP7f |

To comprehensively test the predictive capability of the models, we used five different datasets from different labs (Table 1). Most recordings were collected during task-free spontaneous behavior, though the Ahrens zebrafish dataset involves responses to visual stimuli [11]. Details of the segmentation pipelines used to extract fluorescence traces are described in the original dataset publications. We z-scored all fluorescence traces to zero mean and unit variance. For experiments involving predicting PCs, we computed PCs after z-scoring individual neurons, and the magnitudes of PCs were preserved. We first cut each session into 1K-step segments, then partitioned each segment into training, validation, and test sets by 3:1:1. See the Appendix for more details.

## 4.2 Baselines

We compared POCO against a diverse set of baselines, from basic linear and auto-regressive models to state-of-the-art methods for time-series forecasting and dynamical system reconstruction. (1) **MLP** is POCO model without conditioning (Equation 2). For multi-session modeling, we use a larger MLP (**MLP_L**) with a similar parameter count as POCO. (2) **NLinear, DLinear**[55] are variants of univariate linear models that use a linear projection from the context to predictions. (3) **Latent_PLRNN** uses piece-wise linear RNN to model underlying dynamical states linked to observations through linear projection [24, 35]. (4) **TSMixer** [16] is an all-MLP architecture for TSF based on mixing modules for the time and feature dimensions [16]. (5) **TexFilter** [54] learns context-dependent frequency filters for time-series processing. (6) **AR_Transformer** is a basic autoregressive Transformer [50]. (7) **Netformer** [31] infers inter-neuron connection strength via an attention layer, learning a dynamic interaction graph. Here, we add softmax to attention weights for more stable training for multi-step prediction. (8) **TCN** denotes ModernTCN [33], a recent multivariate pure-convolution architecture for TSF. More details about architectures and training can be found in the Appendix.

## 4.3 Copy Baseline and Prediction Score

Lastly, we considered a naive baseline that copies the last observation

$$f_{copy}\left(\mathbf{x}_{t-C:t}^{(i)}, i\right) = [\mathbf{x}_{t-1}^{(i)}, \mathbf{x}_{t-1}^{(i)}, ...], \text{ (repeat for } P = 16 \text{ steps).} \tag{6}$$

Although extremely simple, due to the slow dynamics of calcium traces, $f_{copy}$ is a strong baseline, especially in short-term forecasting [44]. As a more intuitive metric than the raw MSE loss, we define the prediction score as the relative performance improvement compared to the copy baseline, i.e.,

$$\text{Prediction Score } (f_{model}) = 1 - L(f_{model})/L(f_{copy}), \tag{7}$$

which is similar to $R^2$, but we use the last time step to replace the sample mean.

## 5 Results

Table 2: *POCO achieves highest prediction scores across species and datasets.* Prediction scores across five datasets show that POCO consistently outperforms baselines, especially in the multi-session setting. Zebrafish data are reduced to 512 PCs. 95% CI from 4 seeds.

| Model | Zebrafish, 512 PCs Deisseroth | Ahrens | Mice | C-elegans Zimmer | Flavell |
|---|---|---|---|---|---|
| *Single-Session Models* | | | | | |
| **POCO** | 0.466 ±0.019 | 0.433 ±0.008 | 0.415 ±0.001 | 0.329 ±0.009 | 0.079 ±0.017 |
| MLP | 0.399 ±0.001 | 0.388 ±0.001 | 0.409 ±0.000 | 0.336 ±0.001 | 0.236 ±0.002 |
| NLinear | 0.167 ±0.000 | 0.211 ±0.000 | 0.348 ±0.000 | 0.250 ±0.000 | 0.217 ±0.001 |
| Latent_PLRNN | 0.064 ±0.024 | 0.212 ±0.002 | 0.335 ±0.001 | 0.143 ±0.015 | 0.170 ±0.005 |
| TexFilter | 0.419 ±0.006 | 0.378 ±0.003 | 0.389 ±0.000 | 0.333 ±0.005 | 0.230 ±0.002 |
| NetFormer | 0.204 ±0.013 | 0.208 ±0.008 | 0.329 ±0.000 | 0.145 ±0.004 | 0.168 ±0.002 |
| AR_Transformer | -0.875 ±0.027 | -0.054 ±0.005 | 0.312 ±0.005 | -0.320 ±0.063 | -1.024 ±0.038 |
| DLinear | 0.211 ±0.000 | 0.290 ±0.000 | 0.394 ±0.000 | 0.267 ±0.000 | 0.221 ±0.000 |
| TCN | 0.153 ±0.010 | 0.240 ±0.004 | 0.360 ±0.000 | 0.305 ±0.003 | 0.226 ±0.004 |
| TSMixer | -0.550 ±0.036 | 0.016 ±0.009 | 0.390 ±0.001 | 0.129 ±0.012 | -0.199 ±0.039 |
| *Multi-Session Models* | | | | | |
| **MS_POCO** | **0.525 ±0.004** | **0.440 ±0.003** | **0.420 ±0.002** | **0.364 ±0.005** | 0.213 ±0.030 |
| MS_MLP_L | 0.466 ±0.001 | 0.418 ±0.001 | 0.412 ±0.000 | 0.360 ±0.001 | **0.276 ±0.007** |
| MS_NLinear | 0.165 ±0.000 | 0.202 ±0.000 | 0.347 ±0.000 | 0.253 ±0.000 | 0.221 ±0.000 |
| MS_Latent_PLRNN | 0.149 ±0.002 | 0.248 ±0.007 | 0.355 ±0.000 | 0.118 ±0.011 | 0.183 ±0.006 |
| MS_TexFilter | 0.440 ±0.005 | 0.349 ±0.000 | 0.389 ±0.000 | 0.346 ±0.000 | 0.256 ±0.002 |
| MS_NetFormer | 0.221 ±0.008 | 0.220 ±0.002 | 0.331 ±0.000 | 0.150 ±0.002 | 0.217 ±0.001 |
| MS_AR_Transformer | -0.777 ±0.005 | 0.002 ±0.012 | 0.317 ±0.003 | -0.333 ±0.043 | -0.675 ±0.022 |

Table 3: *Predicting at single-cell resolution in zebrafish.* Prediction scores on raw neural traces (not PCA-reduced) from zebrafish datasets. All models are multi-session. 95% CI from 4 seeds.

| Model | Zebrafish (Ahrens) | Zebrafish (Deisseroth) |
|---|---|---|
| **MS_POCO** | **0.429** ±0.003 | 0.251 ±0.004 |
| MS_MLP_L | 0.423 ±0.001 | **0.264** ±0.000 |
| MS_NLinear | 0.367 ±0.000 | 0.172 ±0.001 |
| MS_TexFilter | 0.398 ±0.001 | 0.232 ±0.004 |

**Multi-Session POCO Outperforms Baselines.** We tested POCO against baselines on five calcium imaging datasets (Table 2). Sample prediction traces for POCO are shown in Figure 2C. For zebrafish datasets, we first considered the more tractable problem of predicting the first 512 principal components (PCs) due to the large number of neurons. We compared training a different model for each session (single-session models) and training a shared model for all sessions (multi-session models, MS). POCO consistently benefited from multi-session training, outperforming all baselines on four out of five datasets. Other models—such as PLRNN, TexFilter, and NetFormer—also show performance gains from multi-session training. We obtained similar results measuring prediction errors by MSE and MAE (mean absolute error), as shown in Table S6 and S7.

We further evaluated a subset of efficient multi-session models on two zebrafish datasets at single-cell resolution (Table 3). Multi-session POCO achieved the best performance on Ahrens' dataset and near-best on Deisseroth's, indicating its ability to model both raw neural activity and PCA-reduced signals. Nonetheless, its advantage over other models was smaller than in PC space, highlighting the difficulty of modeling large neuronal populations with sparse, noisy recordings.

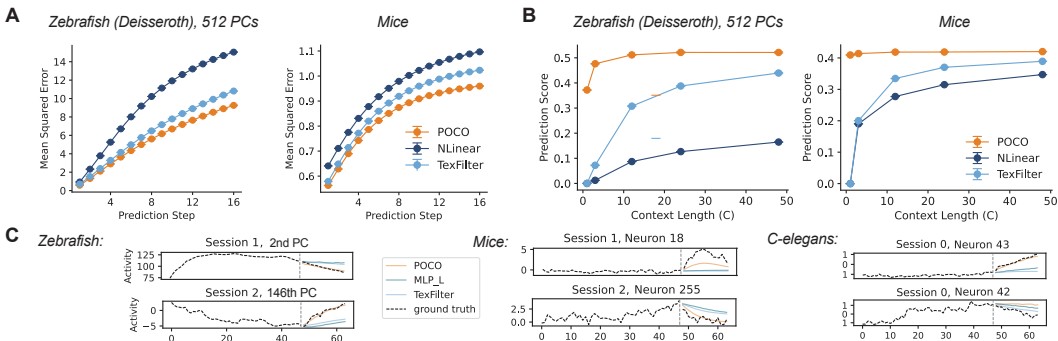

Figure 2: *POCO maintains accuracy advantage over time and benefits from longer context.* (A) MSE increases when forecasting longer into the future. Results are shown for two different datasets, see Figure S6 for additional datasets. Error bars show SEM of 3 random seeds. (B) Model performance improves with longer context. (C) Sample prediction traces produced by POCO, large MLP, and Texfilter. The first $C = 48$ steps are given to the model as context.

**Effect of Context and Prediction Length.** We observed that prediction error gradually increases over $P = 16$ prediction steps, but POCO generated relatively accurate predictions across all time steps (Figure 2A). To evaluate the impact of context length, we also tested on shorter context lengths $C \in \{1, 3, 12, 24\}$, and adjusted POCO's token length $T_C$ to $\{1, 1, 3, 6\}$ to control the number of tokens. We found that univariate models perform poorly with short contexts, consistent with results in recent work [32]. POCO outperformed baselines in different context lengths (Figure 2B). The prediction accuracy increased with context length but plateaus beyond $C > 12$, suggesting that most of the predictive information is contained within a relatively short temporal window.

**POCO Performance Improvements Scale with Recording Length.** We next tested how POCO performance scales with dataset size. First, instead of using the full training partition in each session, we tested the model performance when only the first $T_{train}$ steps are used (Figure 3A, S7). We found that POCO shows steady improvement when longer recordings are used for training, whereas TexFilter shows slower improvements, and NLinear shows no apparent improvement. Second, we split sessions in one dataset into several approximately even partitions and train one model for each partition (Figure 3B, S8). We found that POCO consistently benefits from training on more sessions. Taken together, these results suggest that POCO effectively leverages longer recordings across sessions to learn complex neural dynamics.

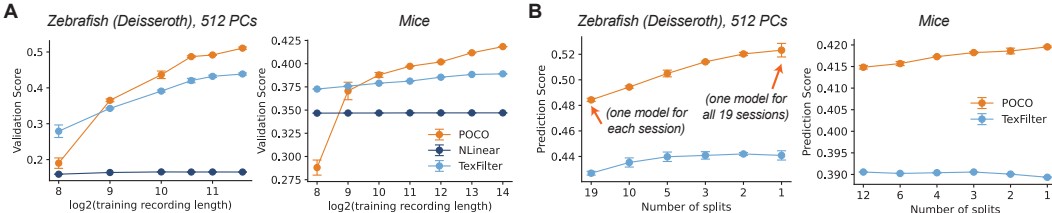

Figure 3: *POCO performance improves with longer recordings and more sessions.* (A) Prediction score vs. training recording length (x-axis in log scale) for two different datasets. Models were trained using increasing portions of each session's data. Error bars show SEM across 3 random seeds. (B) We split all sessions in one dataset into $n$ approximately equal partitions and train one model on each partition, then take the average of model prediction scores across all partitions. Average prediction scores vs the number of splits $n$ is shown for two datasets.

Table 4: *POCO benefits from multi-session, but not multi-species, training.* Comparing single-session, multi-session, multi-species, and zebrafish-only POCO variants. Within-species training yields the best performance. 95% CI from 4 seeds.

| | Zebrafish, 512 PCs | | Mice | C-elegans | |
| Model | Deisseroth | Ahrens | | Zimmer | Flavell |
|---|---|---|---|---|---|
| Single-Session POCO | 0.466 ±0.019 | 0.433 ±0.008 | 0.415 ±0.001 | 0.329 ±0.009 | 0.079 ±0.017 |
| Multi-Session POCO | **0.525** ±0.004 | **0.440** ±0.003 | **0.420** ±0.002 | **0.364** ±0.005 | 0.213 ±0.030 |
| Multi-Species POCO | 0.499 ±0.003 | **0.441** ±0.003 | 0.403 ±0.000 | 0.330 ±0.009 | **0.252** ±0.011 |
| Zebrafish POCO | 0.500 ±0.005 | **0.442** ±0.004 | | | |

**POCO Does Not Consistently Benefit from Multi-Species Training.** To see if datasets from multiple species improves performance, we trained POCO on different datasets simultaneously. Specifically, in each model update step, we aggregated the loss computed from one random batch of data in each dataset. We found that the model generally does not benefit from multi-species training (Table 4). This may be due to differences across datasets in pre-processing pipelines, recording conditions, and, perhaps more importantly, differences in the underlying neural dynamics of recorded species, animals' behavioral states, and specific brain regions (Table 1). Given this result and a recent report that pre-training on other specimens does not improve neural forecasting performance in zebrafish [20], we hypothesize that the model performance only significantly benefits from more sessions when the modeled systems are *similar enough*. We explored this hypothesis in the following simulation experiment.

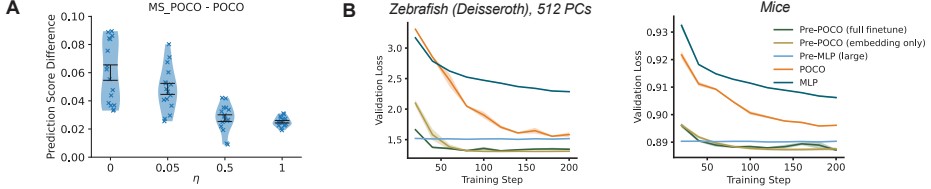

Figure 4: *Multi-session POCO improves when individuals are similar; POCO can quickly adapt to new sessions* (A) Performance gain of multi-session POCO compared to single-session POCO on synthetic data for different values of $\eta$. Larger $\eta$ means individuals are less similar. We randomly generated 16 cohorts, each with 16 individuals. Each blue cross represents a cohort. Error bars are SEM. (B) Validation loss curve of fine-tuning pre-trained POCO (Pre-POCO), pre-trained MLP (Pre-MLP), and training POCO, MLP from scratch. We also compared full-finetuning with only tuning the embedding. Error shades represent SEM for 3 random seeds. Two sample sessions from two datasets are shown here; see Figure S12 for more sessions.

**Simulation.** Here, we used simulated data to test how similarity between individuals influences multi-session model performance. To generate neural data from a synthesized cohort, we first randomly sample a template connectivity matrix $J_0 \sim \mathcal{N}(0, I) \in \mathbb{R}^{n \times n}$, where we use $n = 300$

neurons. Then for each synthesized individual $i \in [16]$, we set

$$J_i = \sqrt{1 - \eta^2} J_0 + \eta \epsilon_i, \ \ \epsilon_i \sim \mathcal{N}(0, I),$$

where $\epsilon_i$ is random deviation from the template connectivity matrix, $\eta \in [0, 1]$ controls the similarity between individuals. We set the coefficient for $J_0$ to be $\sqrt{1 - \eta^2}$ so that $J_i$ keeps unit variance. We then used each $J_i$ as the connectivity of a noisy spontaneous RNN to generate synthetic neural data (see the Appendix for details) and trained POCO as on the multi-session neural data above (Figure 4A). POCO showed greater benefit from multi-session training when individuals shared similar connectivity patterns ($\eta \leq 0.05$), compared to when their connectivity was independent ($\eta = 1$).

We also compared POCO with baselines on single-session simulation data. Surprisingly, models including PLRNN, auto-regressive Transformer, and TSMixer performed significantly better than POCO, despite their relatively poor performance on real neural datasets (Figure S11). Real neural data likely exhibits greater non-stationarity, multi-scale dependencies, and heterogeneous noise profiles than our current simulations, properties which POCO's architecture may be better suited to handle than models excelling in the simulated regime.

**Finetuning on New Sessions.** A core benefit of the foundation model approach is that it allows rapid adaptation to new sessions. We pre-trained POCO on 80% of sessions and fine-tuned this model on each of the remaining ones (see the Appendix for details). For finetuning, we compared full finetuning with only finetuning the unit and session embedding. We found that pre-trained models achieved reasonable predictive performance in only tens of training steps (Figure 4B). In addition, full finetuning leads to limited or no improvements compared to only fine-tuning the unit embedding (Table S9). Rapid adaptation is crucial for real-time, closed-loop applications: in our setup, fine-tuning the embedding for 200 training steps takes less than 15 seconds, and the forecasting inference time is only 3.5 ms (see the Appendix for details). Consistent with previous results on multi-dataset training, pre-training on different datasets does not improve performance (Table S10).

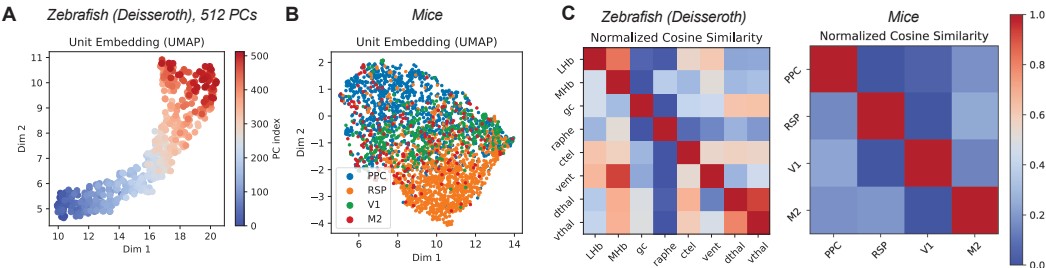

Figure 5: *POCO learns meaningful unit embeddings without supervision.* (A) UMAP [34] visualization of unit embeddings after training POCO on the first 512 PCs in a zebrafish dataset. (B) Visualization of unit embedding after training POCO on the mice dataset, where neurons are colored by the brain region. One sample session is shown for (A) and (B), see Figure S9 for more sessions. (C) Normalized average cosine similarity of unit embeddings between each pair of regions. Each row is normalized to $[0, 1]$ and then averaged across 4 runs. Patterns are consistent for different seeds (Figure S10). See the Appendix for more details.

**Analyzing Unit Embedding.** Recent work shows that unit embeddings in POYO learns region and cell-type-specific information when trained for behavioral decoding [6]. We analyzed the trained multi-session POCO model to determine if similar structure emerges. We found that when trained on PCs, unit embeddings are consistently distributed according to the order of the PCs (Figure 5A). When trained on neurons, we found significant clustering of neurons by brain region, as indicated by the average pair-wise cosine similarity (Figure 5C). In particular, retrosplenial cortex (RSP) neurons in mice form a particularly distinct cluster (Figure 5B). Thus, POCO learns to encode functional and dynamical properties in the unit embedding when trained for forecasting, even when no prior knowledge (e.g., neuron location) is given to the model.

**Effect of Filters.** Filtering high or low frequency components is a common preprocessing technique in calcium imaging, used to remove slow drifts or fast noise that are not directly related to neural dynamics [23, 36]. By default, we used no temporal filter to maximally preserve neural signals. However, we found that with low-filtering, POCO still outperforms baselines except on *C. elegans*

datasets (Table S8). Low-pass filters improved models' performance compared to the copy baseline in most datasets, suggesting that high-frequency components are generally harder to predict (Figure S13). Low-pass filtering also helped POCO to benefit more from multi-session training (Figure S8).

**Zapbench Evaluation.** Although the main focus of this work is on multi-session datasets, we also tested our model on a recent neural population forecasting benchmark, Zapbench [32], which contains light-sheet microscopy recordings of 71721 neurons over 7879 time steps for one zebrafish. We followed the setup in Zapbench to partition the dataset and evaluate our model with short ($C = 4$) or long ($C = 256$) context, and prediction length $P = 32$. We found that when $C = 4$, POCO outperforms other trace-based methods and performs comparably to UNet, a computationally expensive model operating directly on raw volumetric videos rather than segmented neural traces [20]. At longer contexts ($C = 256$), POCO underperforms relative to UNet and TSMixer (Figure S14). See the Appendix for more details.

Table 5: *Both MLP forecaster and population encoder are necessary for POCO.* Ablation study shows performance drops when either component is removed or simplified. 95% CI from 4 seeds.

| Model | Zebrafish, 512 PCs | | Mice |
| | Deisseroth | Ahrens | |
| --- | --- | --- | --- |
| **Full POCO** | **0.525** ±0.004 | **0.440** ±0.003 | **0.420** ±0.002 |
| POYO only | -0.971 ±0.015 | -0.057 ±0.001 | 0.332 ±0.001 |
| MLP only | 0.417 ±0.002 | 0.370 ±0.001 | 0.409 ±0.000 |
| MLP conditioned by univariate Transformer | 0.463 ±0.001 | 0.408 ±0.002 | 0.411 ±0.000 |
| MLP conditioned by unit embedding | 0.481 ±0.004 | 0.434 ±0.004 | 0.413 ±0.000 |

**Ablation Study.** To test the necessity of the components of POCO, we removed or replaced some parts of the model and compared performance. Specifically, we tested (1) directly using the POYO model to generate 16-steps prediction instead of generating conditioning parameters (POYO only); (2) MLP without conditioning; (3) MLP conditioned by a univariate Transformer that takes the calcium trace of a single neuron instead of encoding the whole population; (4) 3-layer MLP directly conditioned by unit embedding (see the Appendix for more details). Both the MLP forecaster and the population encoder were necessary for full performance (Table 5). We also tested how key model hyperparameters influence performance, including the length of the token (Figure S15), embedding dimension (Figure S16), number of layers (Figure S17), the number of latents (Figure S18), learning rate (Figure S19) and weight decay (Figure S20). We found that POCO performance is relatively stable for a range of hyperparameter settings, and even a small POYO encoder is sufficient.

## 6 Discussion

In this work, we introduced POCO, a population-conditioned forecaster that combines a local univariate predictor with a global population encoder to capture both neuron-level dynamics and shared brain-state structure. Across five calcium-imaging datasets in zebrafish, mice, and *C. elegans*, POCO achieves state-of-the-art overall forecasting accuracy. Beyond raw performance, we show that POCO rapidly adapts to new sessions with only tens of fine-tuning steps of its embeddings, making it feasible for real-time adaptation during live recordings. Our analysis of POCO's learned unit embeddings demonstrates that the model autonomously uncovers meaningful population structure such as brain regions, even though no anatomical labels were provided. These findings underscore POCO's dual strength in accurate prediction and in learning interpretable representations of units. Finally, our results extend the recent progress on scaling up models for behavioral decoding [5, 6] to the realm of neural prediction on spontaneous neural recordings in different species.

We note a few limitations that can be opportunities for future research. First, our results indicate that factors such as calcium indicator dynamics, preprocessing pipelines, and species differences can significantly affect model performance, yet a systematic understanding of their influence remains lacking (see Appendix for more additional discussion). Second, our findings highlight the difficulty of multi-dataset and multi-species training—a challenge that may be mitigated by improved architectures or alignment strategies. Third, while POCO learns biologically meaningful unit embeddings within datasets, it is unclear whether these representations are comparable across species or generalize to unseen brain regions. Finally, while we focus on calcium imaging during spontaneous behavior, extending POCO to spiking data and standardized behavioral tasks could enable the use of larger datasets and support modeling of neural dynamics in more structured settings [14, 26].

## Acknowledgments and Disclosure of Funding

This work was supported by the NIH (RF1DA056403), James S. McDonnell Foundation (220020466), Simons Foundation (Pilot Extension-00003332-02), McKnight Endowment Fund, CIFAR Azrieli Global Scholar Program, and NSF (2046583).

We would like to thank Sabera Talukder, Krystal Xuejing Pan, and Viren Jain for helpful discussions.

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

# A Supplementary Results

Table S6: **MSE comparison across datasets.** Mean squared error (MSE) across five datasets, using the same model configurations as in Table 2. Different from Table 2 where the score is computed by averaging across all sessions, here sessions are weighted by the size of the test set ($N_i \times |D_{test}|$, where $D_{test}$ is the number of sequences in the test set), so longer recording sessions and sessions with more neurons recorded will receive a larger weight.

| Model | Zebrafish, 512 PCs Deisseroth | Ahrens | Mice | C-elegans Zimmer | Flavell |
|---|---|---|---|---|---|
| *Single-Session Models* | | | | | |
| **POCO** | 6.022 ±0.083 | **40.118 ±0.473** | 0.845 ±0.002 | 0.369 ±0.005 | 0.696 ±0.011 |
| MLP | 7.082 ±0.021 | 43.335 ±0.024 | 0.853 ±0.000 | 0.365 ±0.001 | 0.582 ±0.001 |
| NLinear | 9.587 ±0.005 | 55.742 ±0.016 | 0.944 ±0.000 | 0.411 ±0.000 | 0.597 ±0.000 |
| Latent_PLRNN | 10.423 ±0.152 | 55.047 ±0.189 | 0.964 ±0.001 | 0.470 ±0.007 | 0.633 ±0.005 |
| TexFilter | 6.590 ±0.066 | 44.040 ±0.215 | 0.882 ±0.000 | 0.368 ±0.003 | 0.588 ±0.001 |
| NetFormer | 8.877 ±0.129 | 56.169 ±0.527 | 0.973 ±0.000 | 0.466 ±0.002 | 0.635 ±0.001 |
| AR_Transformer | 18.929 ±0.095 | 73.523 ±0.632 | 1.008 ±0.012 | 0.688 ±0.032 | 1.517 ±0.029 |
| DLinear | 8.986 ±0.003 | 49.993 ±0.018 | 0.876 ±0.000 | 0.401 ±0.000 | 0.594 ±0.000 |
| TCN | 9.575 ±0.076 | 53.725 ±0.254 | 0.926 ±0.001 | 0.383 ±0.002 | 0.591 ±0.003 |
| TSMixer | 15.516 ±0.354 | 68.733 ±0.682 | 0.882 ±0.002 | 0.468 ±0.004 | 0.911 ±0.027 |
| *Multi-Session Models* | | | | | |
| **MS_POCO** | **5.498 ±0.063** | **39.678 ±0.198** | **0.839 ±0.003** | **0.350 ±0.003** | 0.600 ±0.023 |
| MS_MLP_L | 6.108 ±0.011 | 41.294 ±0.110 | 0.850 ±0.000 | 0.352 ±0.000 | **0.552 ±0.005** |
| MS_NLinear | 9.565 ±0.015 | 56.477 ±0.006 | 0.946 ±0.000 | 0.409 ±0.000 | 0.594 ±0.000 |
| MS_Latent_PLRNN | 9.597 ±0.026 | 52.571 ±0.435 | 0.936 ±0.000 | 0.476 ±0.005 | 0.621 ±0.004 |
| MS_TexFilter | 6.377 ±0.053 | 46.221 ±0.055 | 0.882 ±0.001 | 0.361 ±0.000 | 0.568 ±0.001 |
| MS_NetFormer | 8.803 ±0.076 | 55.351 ±0.210 | 0.970 ±0.001 | 0.461 ±0.001 | 0.597 ±0.000 |
| MS_AR_Transformer | 18.268 ±0.025 | 70.082 ±0.767 | 0.997 ±0.004 | 0.696 ±0.015 | 1.261 ±0.017 |

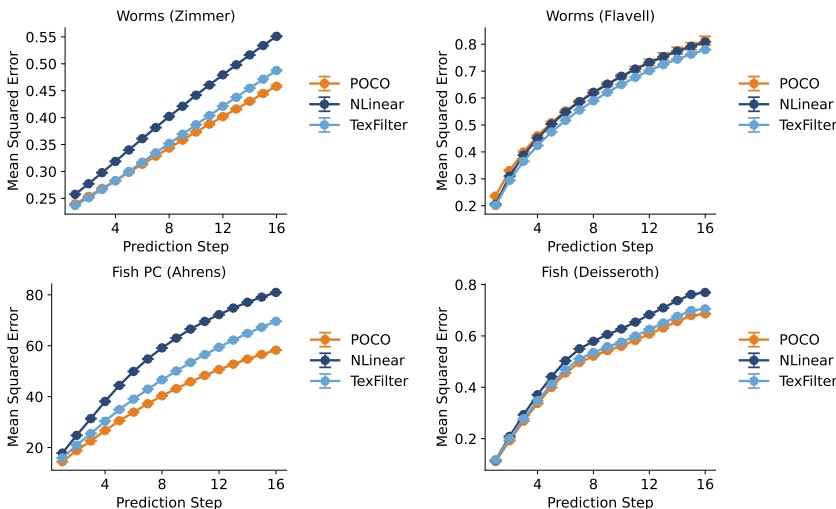

Figure S6: **Forecasting error increases with prediction horizon across datasets.** Mean squared error (MSE) plotted over 16 prediction steps for four additional datasets. POCO maintains better performance over time compared to NLinear and TexFilter. Error bars show SEM across 3 seeds.

Table S7: **Mean absolute error (MAE) comparison across datasets.** Same as Table S6, but reporting mean absolute error (MAE) instead of prediction score. 95% confidence intervals estimated with 4 random seeds.

| Model | Zebrafish, 512 PCs | | Mice | C-elegans | |
| | Deisseroth | Ahrens | | Zimmer | Flavell |
|---|---|---|---|---|---|
| *Single-Session Models* | | | | | |
| **POCO** | 1.310 ±0.005 | 4.116 ±0.006 | 0.645 ±0.001 | 0.418 ±0.004 | 0.604 ±0.007 |
| MLP | 1.442 ±0.002 | 4.322 ±0.001 | 0.648 ±0.000 | 0.413 ±0.001 | 0.540 ±0.001 |
| NLinear | 1.818 ±0.000 | 4.872 ±0.001 | 0.686 ±0.000 | 0.438 ±0.000 | 0.545 ±0.000 |
| Latent_PLRNN | 1.965 ±0.009 | 5.092 ±0.008 | 0.709 ±0.000 | 0.484 ±0.004 | 0.559 ±0.002 |
| TexFilter | 1.443 ±0.010 | 4.416 ±0.012 | 0.657 ±0.000 | 0.409 ±0.002 | 0.537 ±0.001 |
| NetFormer | 1.757 ±0.013 | 4.852 ±0.007 | 0.693 ±0.000 | 0.471 ±0.001 | 0.560 ±0.001 |
| AR_Transformer | 2.465 ±0.010 | 5.500 ±0.007 | 0.712 ±0.006 | 0.606 ±0.014 | 0.941 ±0.009 |
| DLinear | 1.761 ±0.000 | 4.645 ±0.001 | 0.658 ±0.000 | 0.440 ±0.000 | 0.547 ±0.000 |
| TCN | 1.847 ±0.016 | 4.822 ±0.002 | 0.674 ±0.000 | 0.421 ±0.001 | 0.539 ±0.002 |
| TSMixer | 2.026 ±0.039 | 5.021 ±0.035 | 0.665 ±0.001 | 0.493 ±0.001 | 0.702 ±0.010 |
| *Multi-Session Models* | | | | | |
| **MS_POCO** | **1.262 ±0.004** | **4.096 ±0.006** | **0.643 ±0.002** | **0.402 ±0.003** | 0.556 ±0.011 |
| MS_MLP_L | 1.358 ±0.002 | 4.191 ±0.002 | 0.647 ±0.001 | 0.403 ±0.003 | **0.522 ±0.004** |
| MS_NLinear | 1.824 ±0.001 | 4.868 ±0.003 | 0.687 ±0.000 | 0.437 ±0.000 | 0.545 ±0.000 |
| MS_Latent_PLRNN | 1.892 ±0.004 | 4.966 ±0.005 | 0.694 ±0.000 | 0.488 ±0.002 | 0.561 ±0.003 |
| MS_TexFilter | 1.418 ±0.011 | 4.444 ±0.007 | 0.658 ±0.001 | 0.404 ±0.000 | 0.526 ±0.002 |
| MS_NetFormer | 1.766 ±0.006 | 4.849 ±0.004 | 0.692 ±0.000 | 0.469 ±0.001 | 0.544 ±0.000 |
| MS_AR_Transformer | 2.454 ±0.003 | 5.421 ±0.065 | 0.715 ±0.002 | 0.615 ±0.007 | 0.862 ±0.006 |

Table S8: **Low-pass filtering improves prediction in most datasets.** Performance across five datasets after applying a low-pass filter with a cutoff of $0.1 \times f_s$, where $f_s$ is the sampling frequency. Filtering improves POCO's performance relative to the copy baseline in most settings. Results are averaged across sessions; 95% confidence intervals from 4 seeds.

| Model | Zebrafish, 512 PCs | Mice | C-elegans | |
| | Ahrens | | Zimmer | Flavell |
|---|---|---|---|---|
| *Single-Session Models* | | | | |
| **POCO** | 0.60 ± 0.02 | 0.50 ± 0.00 | 0.21 ± 0.02 | -0.04 ± 0.03 |
| MLP | 0.59 ± 0.00 | 0.50 ± 0.00 | 0.31 ± 0.00 | 0.35 ± 0.00 |
| NLinear | 0.33 ± 0.00 | 0.33 ± 0.00 | 0.16 ± 0.00 | 0.27 ± 0.00 |
| Latent_PLRNN | 0.21 ± 0.00 | 0.19 ± 0.00 | 0.13 ± 0.02 | 0.16 ± 0.01 |
| TexFilter | 0.61 ± 0.01 | 0.50 ± 0.01 | 0.37 ± 0.01 | 0.34 ± 0.01 |
| NetFormer | 0.27 ± 0.00 | 0.31 ± 0.00 | 0.01 ± 0.01 | 0.09 ± 0.00 |
| AR_Transformer | -0.20 ± 0.02 | -0.14 ± 0.01 | -1.37 ± 0.13 | -1.48 ± 0.03 |
| DLinear | 0.39 ± 0.00 | 0.41 ± 0.00 | 0.17 ± 0.00 | 0.26 ± 0.00 |
| TCN | 0.32 ± 0.01 | 0.31 ± 0.01 | 0.19 ± 0.03 | 0.21 ± 0.02 |
| TSMixer | -0.05 ± 0.03 | 0.32 ± 0.00 | -0.45 ± 0.03 | -0.46 ± 0.06 |
| *Multi-Session Models* | | | | |
| **MS_POCO** | **0.65 ± 0.01** | **0.55 ± 0.01** | 0.32 ± 0.02 | 0.13 ± 0.05 |
| MS_MLP | 0.57 ± 0.00 | 0.51 ± 0.00 | 0.35 ± 0.00 | 0.40 ± 0.00 |
| MS_NLinear | 0.32 ± 0.00 | 0.35 ± 0.00 | 0.19 ± 0.00 | 0.30 ± 0.00 |
| MS_Latent_PLRNN | 0.24 ± 0.01 | 0.21 ± 0.00 | 0.15 ± 0.01 | 0.19 ± 0.01 |
| MS_TexFilter | 0.60 ± 0.01 | 0.51 ± 0.01 | **0.41 ± 0.01** | **0.42 ± 0.01** |
| MS_NetFormer | 0.31 ± 0.01 | 0.33 ± 0.00 | 0.11 ± 0.00 | 0.23 ± 0.00 |
| MS_AR_Transformer | -0.12 ± 0.01 | 0.01 ± 0.01 | -1.59 ± 0.10 | -1.07 ± 0.02 |

Table S9: **Fine-tuning POCO embeddings enables rapid adaptation.** Test performance of fine-tuning a pre-trained POCO (Pre-POCO) versus training from scratch, and also compared pre-trained MLP and other ablation models. In particular, TMLP denotes an MLP conditioned by a univariate Transformer, and UMLP denotes an MLP conditioned by unit embedding (see Ablation Study section and Appendix for details). For POCO, we compared full fine-tuning, embedding-only tuning, and joint MLP+embedding tuning. Notably, embedding-only tuning achieved performance comparable to full fine-tuning. Reported values are means with 95% confidence intervals over 3 seeds.

| Model | Zebrafish, 512 PCs | | Mice |
| | Deisseroth | Ahrens | |
| --- | --- | --- | --- |
| Pre-POCO (full finetune) | **0.532** ±0.014 | **0.424** ±0.017 | **0.406** ±0.001 |
| Pre-POCO (embedding only) | **0.539** ±0.008 | 0.392 ±0.062 | **0.405** ±0.001 |
| Pre-POCO (unit embedding + MLP) | **0.534** ±0.009 | **0.412** ±0.015 | 0.405 ±0.000 |
| Pre-TMLP | 0.470 ±0.002 | 0.423 ±0.017 | 0.400 ±0.000 |
| Pre-MLP (large) | 0.470 ±0.006 | **0.432** ±0.007 | 0.401 ±0.000 |
| Pre-UMLP (large, full finetune) | 0.495 ±0.003 | 0.427 ±0.013 | 0.404 ±0.000 |
| Pre-UMLP (large, embedding only) | 0.499 ±0.001 | 0.400 ±0.007 | 0.404 ±0.000 |
| POCO | 0.497 ±0.019 | 0.424 ±0.020 | 0.404 ±0.001 |
| NLinear | 0.153 ±0.001 | 0.197 ±0.000 | 0.327 ±0.002 |
| MLP | 0.386 ±0.003 | 0.349 ±0.005 | 0.395 ±0.000 |

Table S10: **Pre-training on mismatched datasets hurts performance.** POCO models pre-trained on one zebrafish dataset and fine-tuned on another perform worse than models trained from scratch on the target dataset. Even joint pre-training on both datasets underperforms single-dataset training. These results highlight the importance of within-distribution pre-training. 95% confidence intervals estimated with 3 seeds.

| Model | Zebrafish, 512 PCs | | Mice |
| | Deisseroth | Ahrens | |
| --- | --- | --- | --- |
| Pre-POCO(Ahrens) | 0.464 ±0.005 | **0.392** ±0.062 | 0.290 ±0.014 |
| Pre-POCO(Deisseroth) | **0.539** ±0.005 | 0.349 ±0.031 | 0.338 ±0.005 |
| Pre-POCO(Both Datasets) | 0.517 ±0.015 | **0.410** ±0.015 | 0.304 ±0.007 |
| POCO | 0.497 ±0.019 | **0.424** ±0.020 | **0.398** ±0.014 |

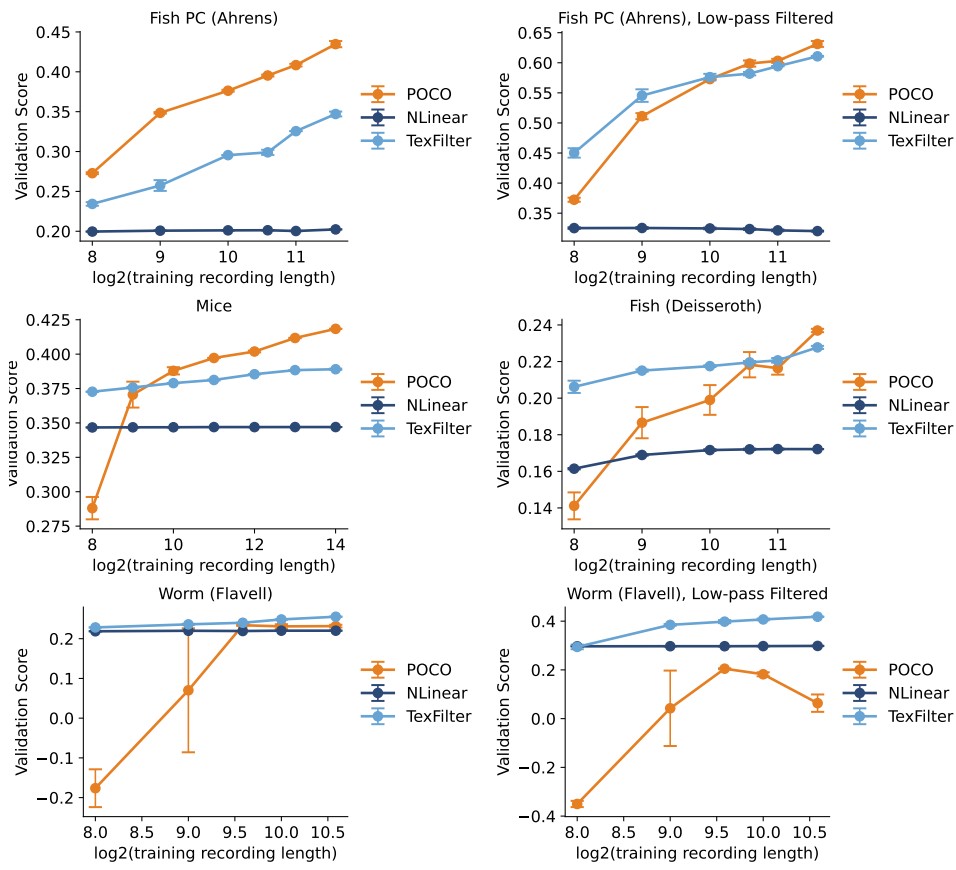

Figure S7: **Longer training recordings improve POCO performance across datasets.** Prediction score increases with log-scaled training duration. Low-pass filtering further enhances gains in some datasets. Error bars show SEM across 2 seeds.

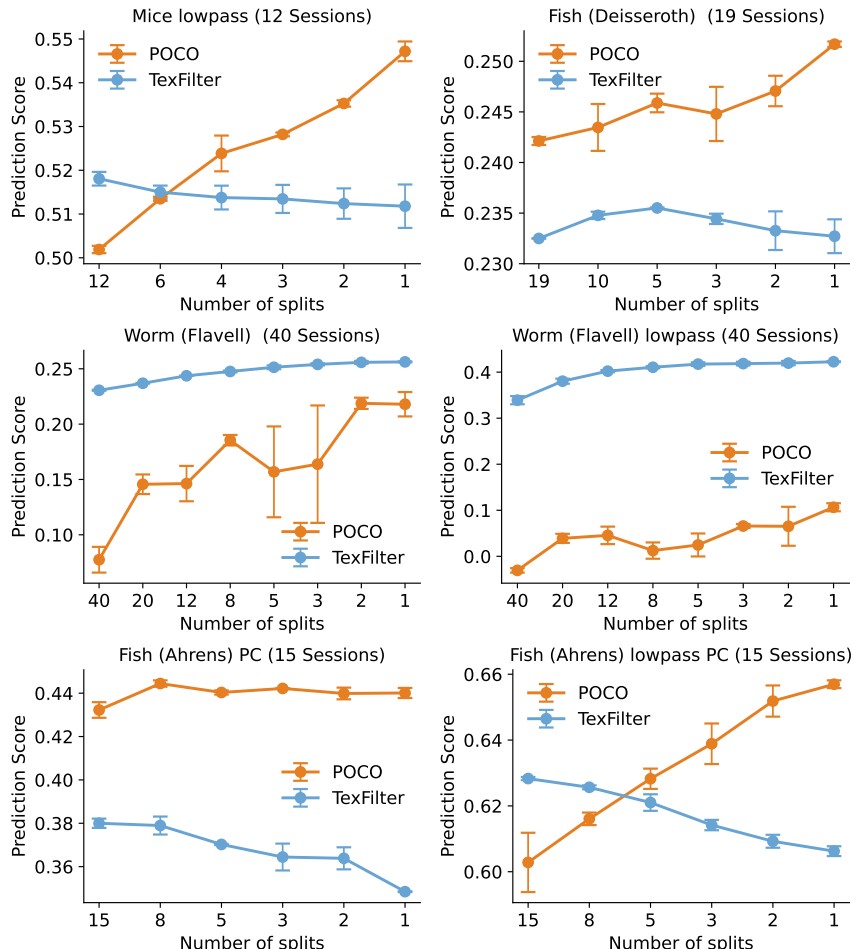

Figure S8: **Multi-session POCO benefits from training on more sessions.** Prediction score improves as more sessions are aggregated. Results are shown for both raw and low-pass filtered datasets. Error bars show SEM across 2 seeds.

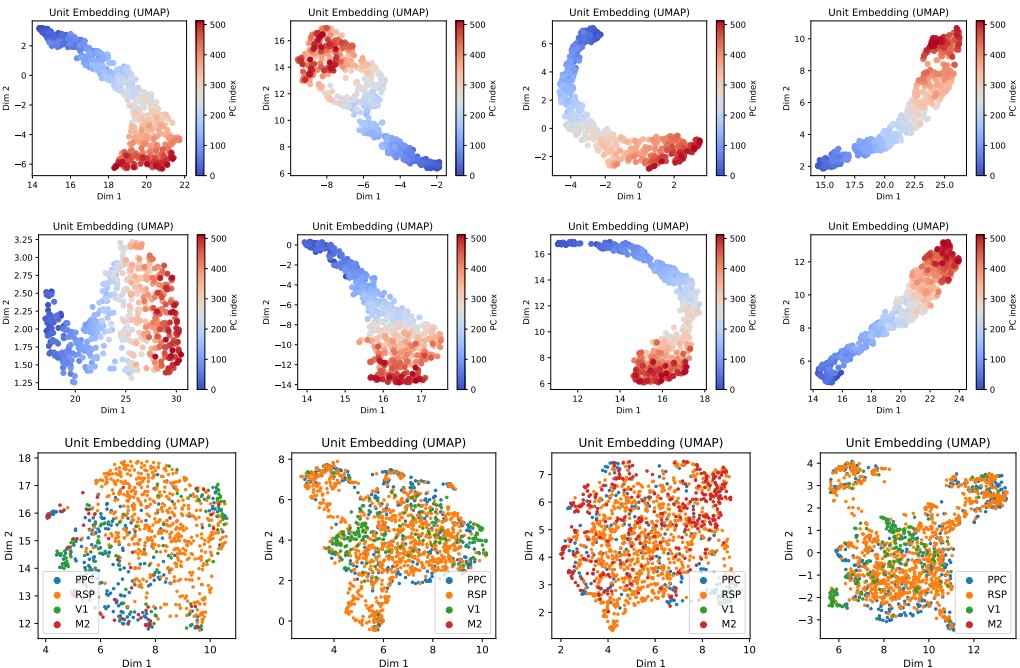

Figure S9: **POCO unit embeddings reflect meaningful structure across sessions.** UMAP projections of unit embeddings from four sessions across zebrafish (Deisseroth and Ahrens) and mouse datasets. Embedding structure is consistent within datasets but varies across sessions.

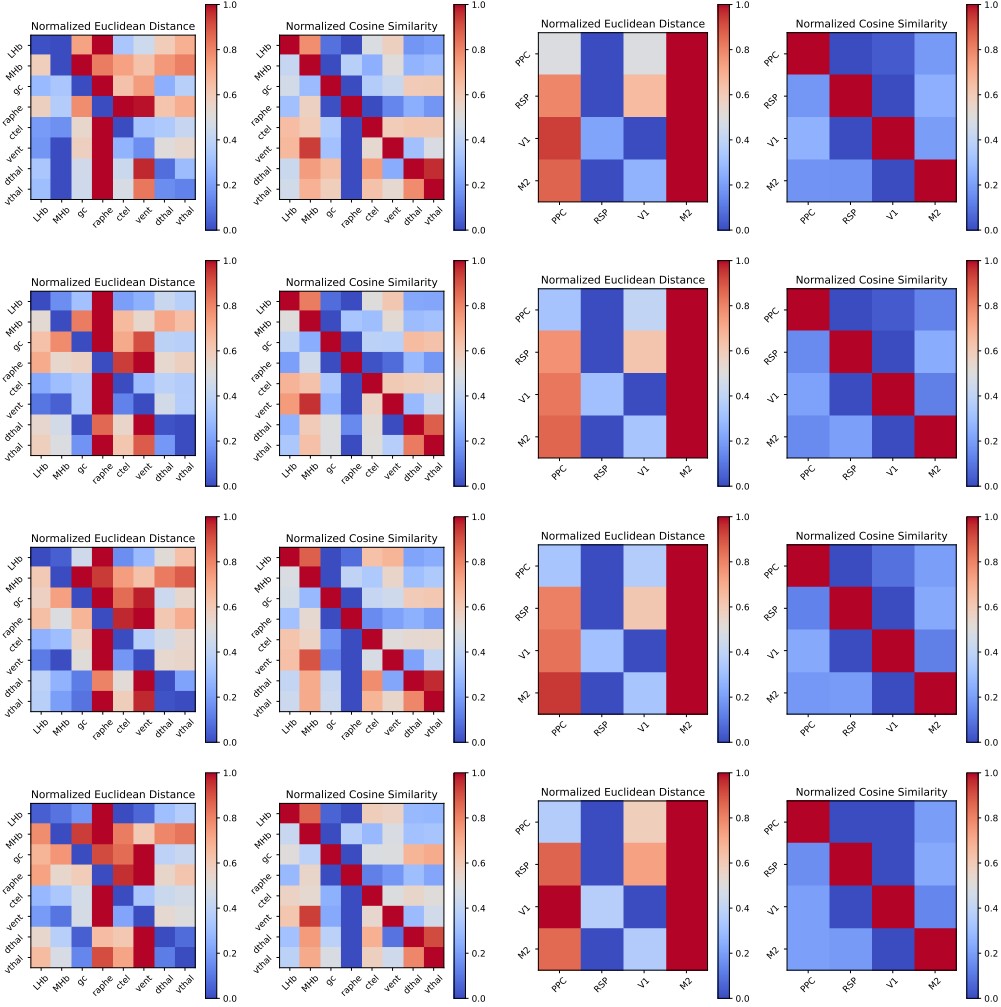

Figure S10: **Unit embeddings cluster by brain region across runs and species.** Normalized cosine similarity and Euclidean distance matrices computed on unit embeddings across zebrafish and mouse sessions. As in Figure 5C, each row is normalized to [0, 1]. From top to bottom are 4 different runs of POCO.

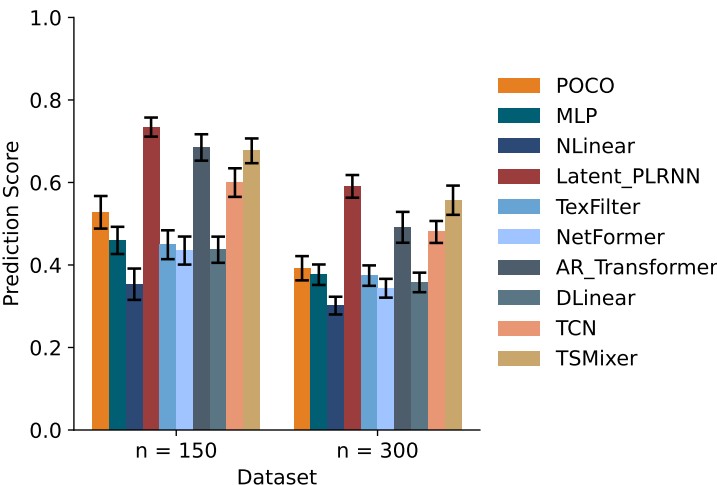

Figure S11: **Baseline models perform well on simulated dynamics.** On single-session synthetic data, PLRNN and TSMixer outperform POCO, unlike on real data. Each point represents a random network instance. Error bars show SEM across 16 seeds.

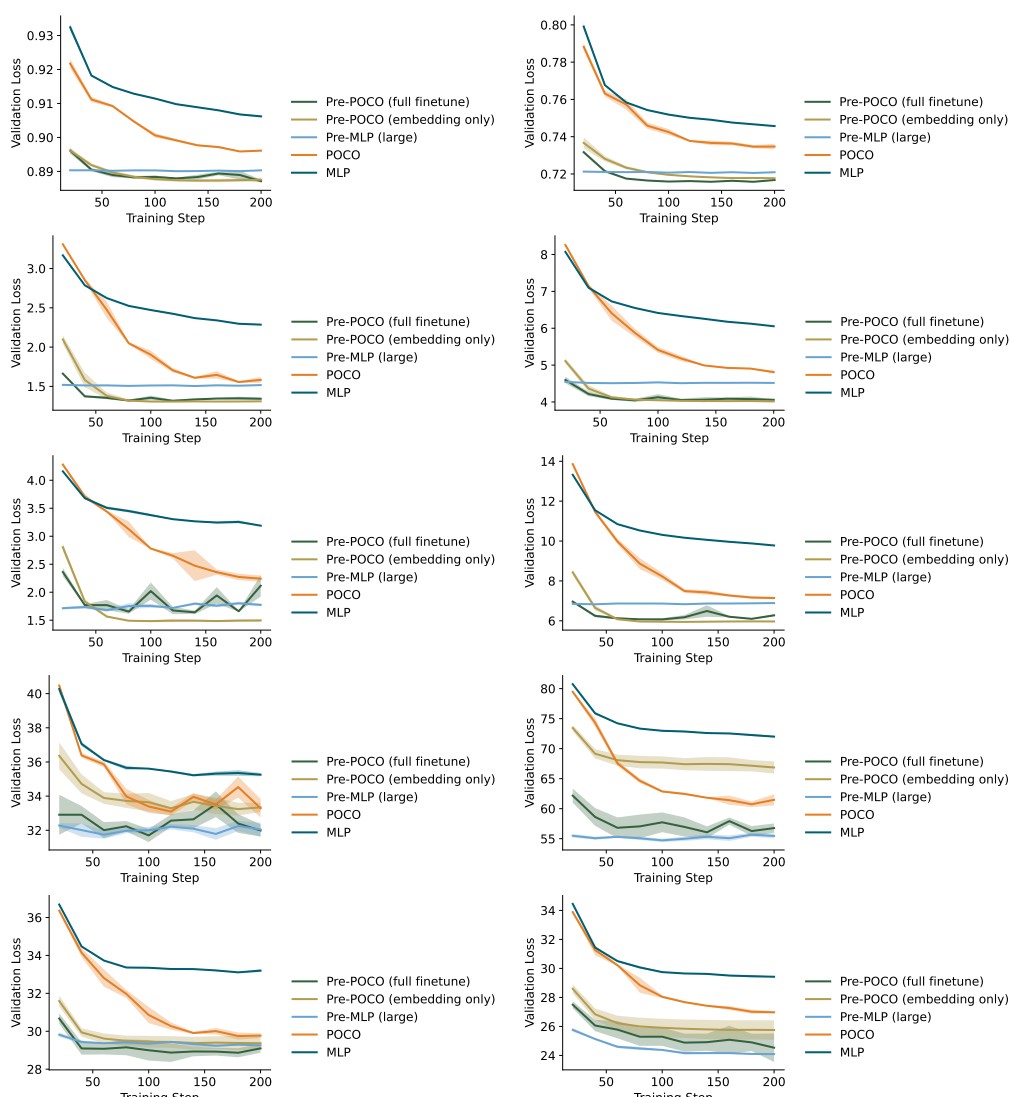

Figure S12: **Fine-tuned POCO rapidly adapts to new sessions with minimal updates.** Validation loss over 200 training steps for various fine-tuning strategies across 10 sessions. The top row contains the results of 2 sessions in the mice dataset. The second and third rows contain results of 4 sessions for Deisseroth's zebrafish dataset, where we are predicting the first 512 PCs. The last 2 rows contain results of 4 sessions for Ahren's zebrafish dataset, where we are also predicting the first 512 PCs. Error shades: SEM across 3 seeds.

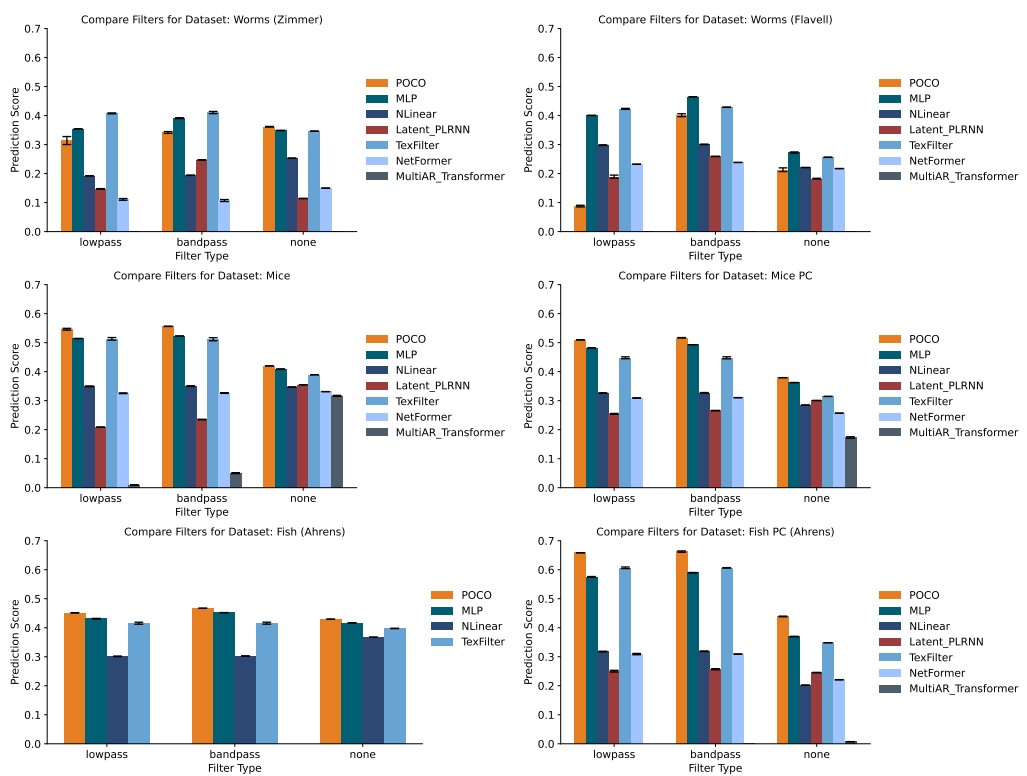

Figure S13: **Low-pass filtering improves POCO forecasting in most datasets.** Prediction score under different filtering regimes (none, low-pass, band-pass) across datasets. Cutoff frequency for low-pass filter is $0.1 \times f_s$, where $f_s$ is the sampling frequency. A band-pass filter additionally removes frequency components lower than $5 \times 10^{-3} \times f_s$. Error bars: SEM over 2 seeds.

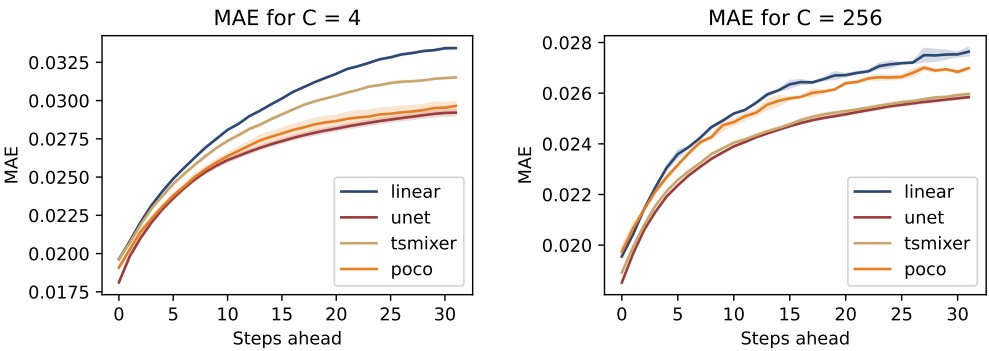

Figure S14: **POCO matches or outperforms volumetric models on Zapbench at short context.** Mean absolute error over 32 prediction steps on Zapbench with $C = 4$ and $C = 256$. POCO matches UNet at short context but underperforms with longer context. Performance data for models other than POCO are from [32].

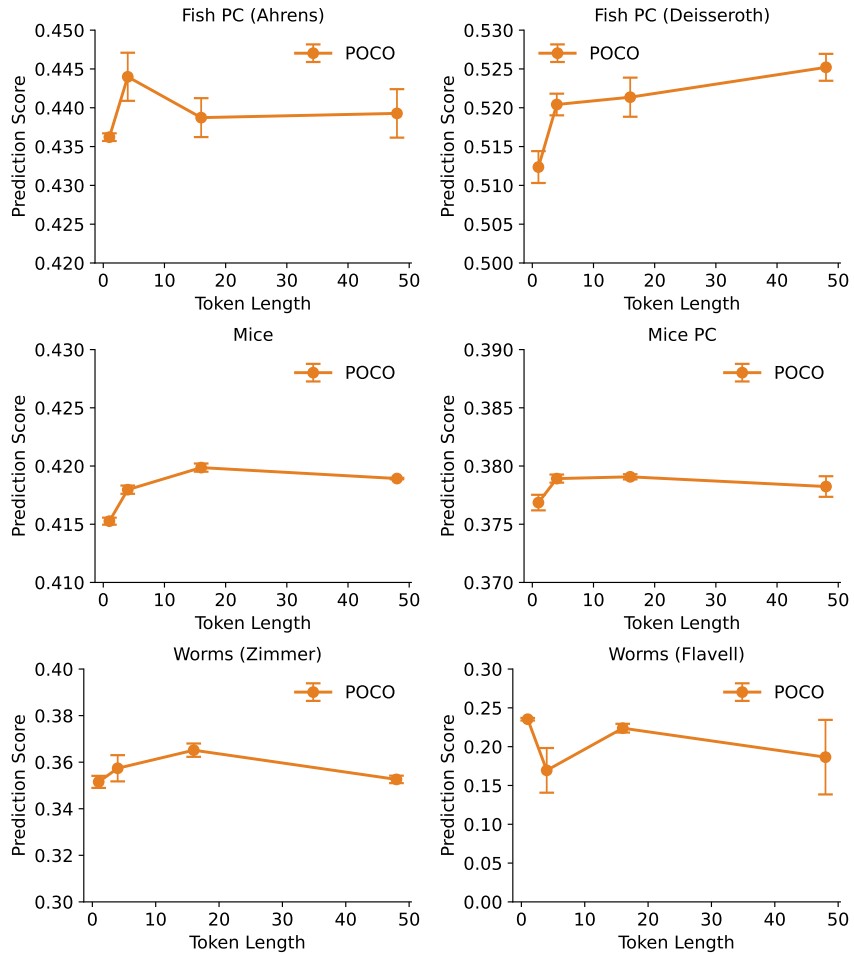

Figure S15: **Token length $T_C$ influences POCO performance across datasets.** Using $T_C = 1$ is suboptimal and computationally expensive. $T_C = 16$ balances performance and cost. Results averaged over 3 seeds.

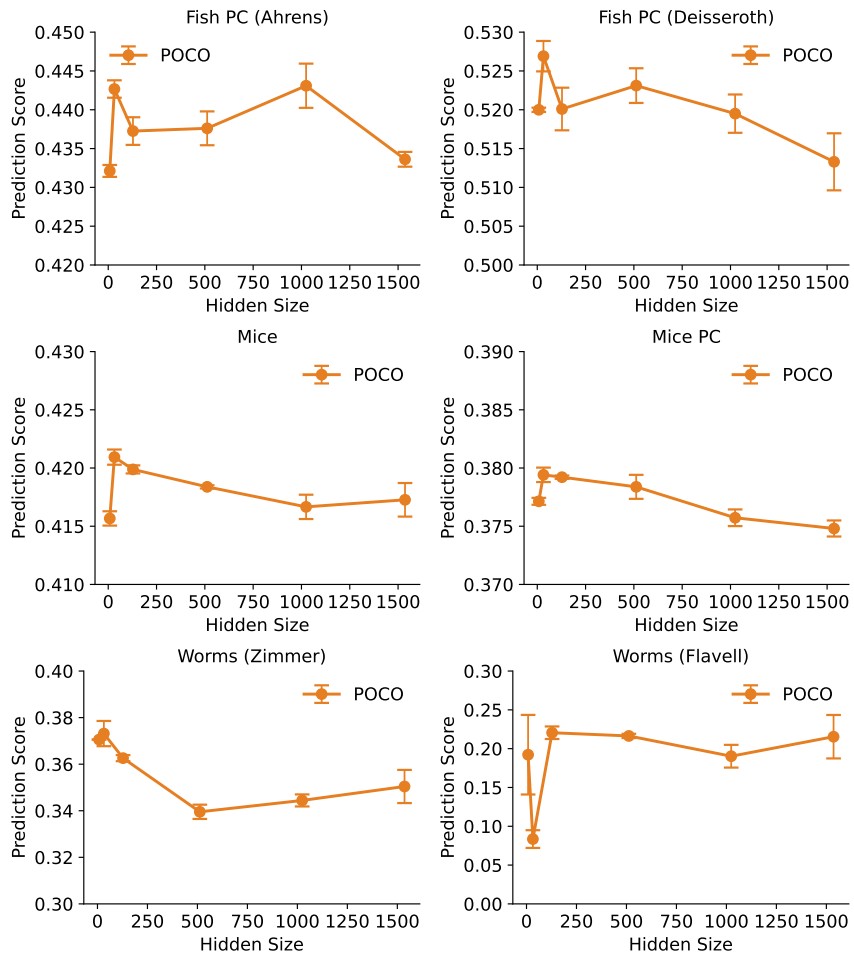

Figure S16: **Embedding size moderately affects POCO performance.** Validation performance shown across 6 datasets as a function of embedding dimension $d$. Default value $d = 128$ achieves near-peak performance. SEM across 3 seeds.

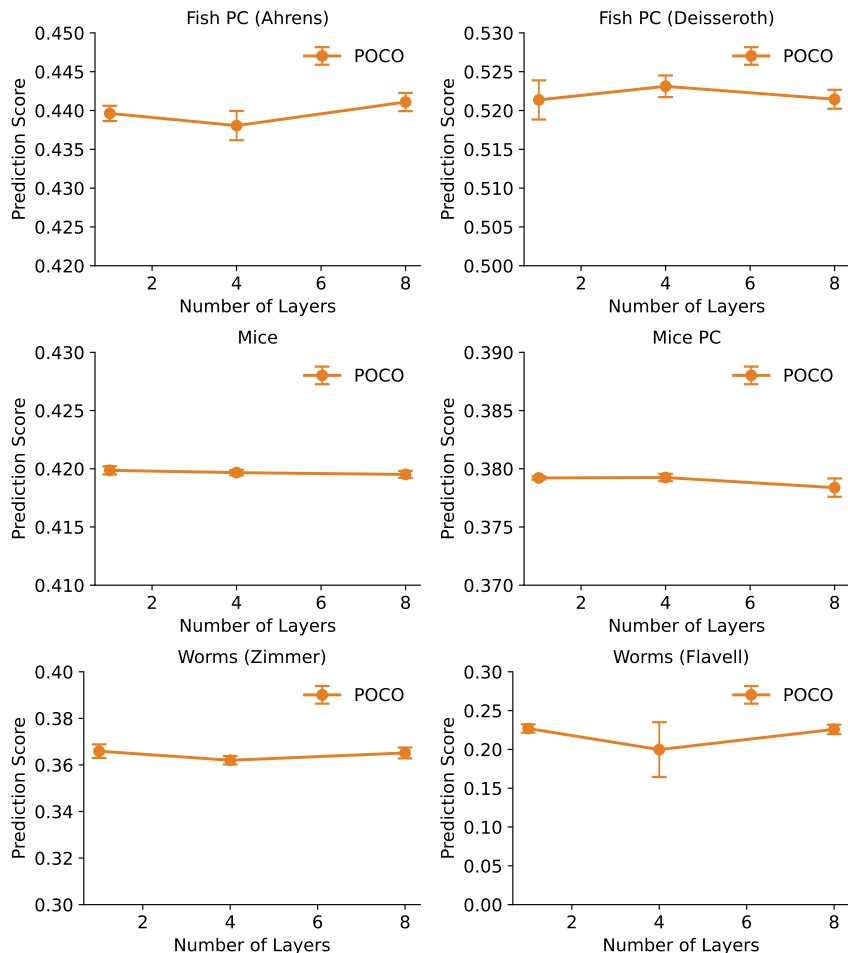

Figure S17: **POCO performs robustly across number of encoder layers.** Increasing Perceiver-IO layers from 1 to 8 shows minimal gain, suggesting 1 layer suffices for most datasets. Error bars: SEM across 3 seeds.

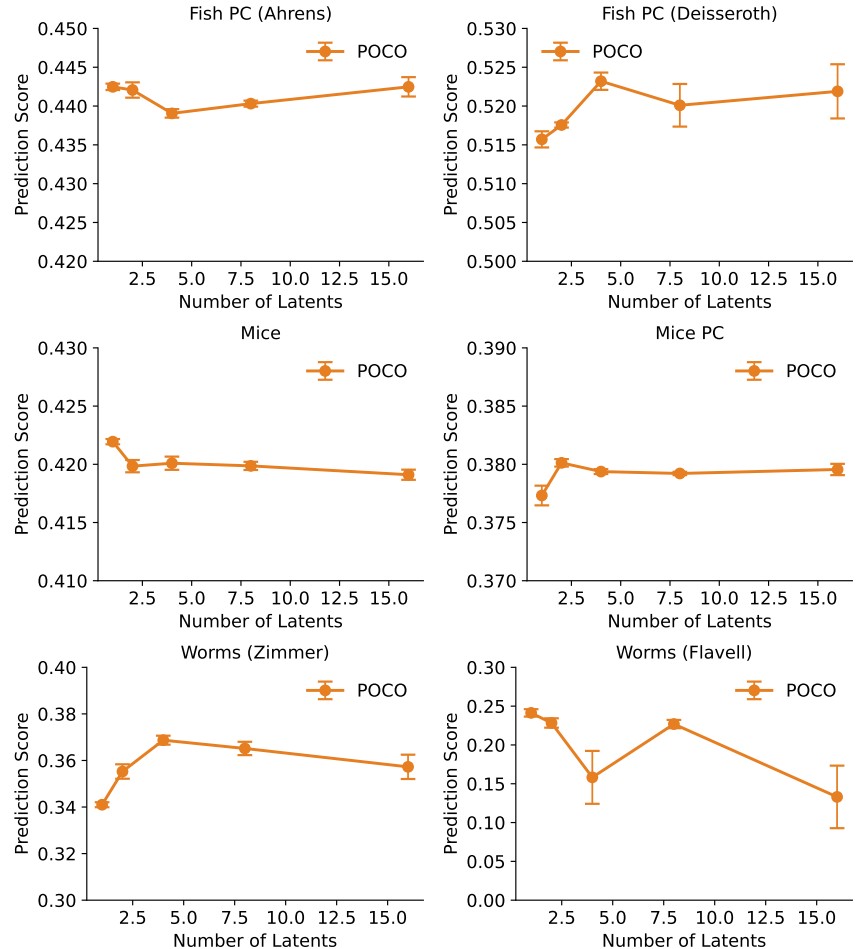

Figure S18: **Number of latents affects POCO performance marginally.** Validation score shown for different latent counts in the encoder. Performance saturates near 8 latents. SEM across 3 seeds.

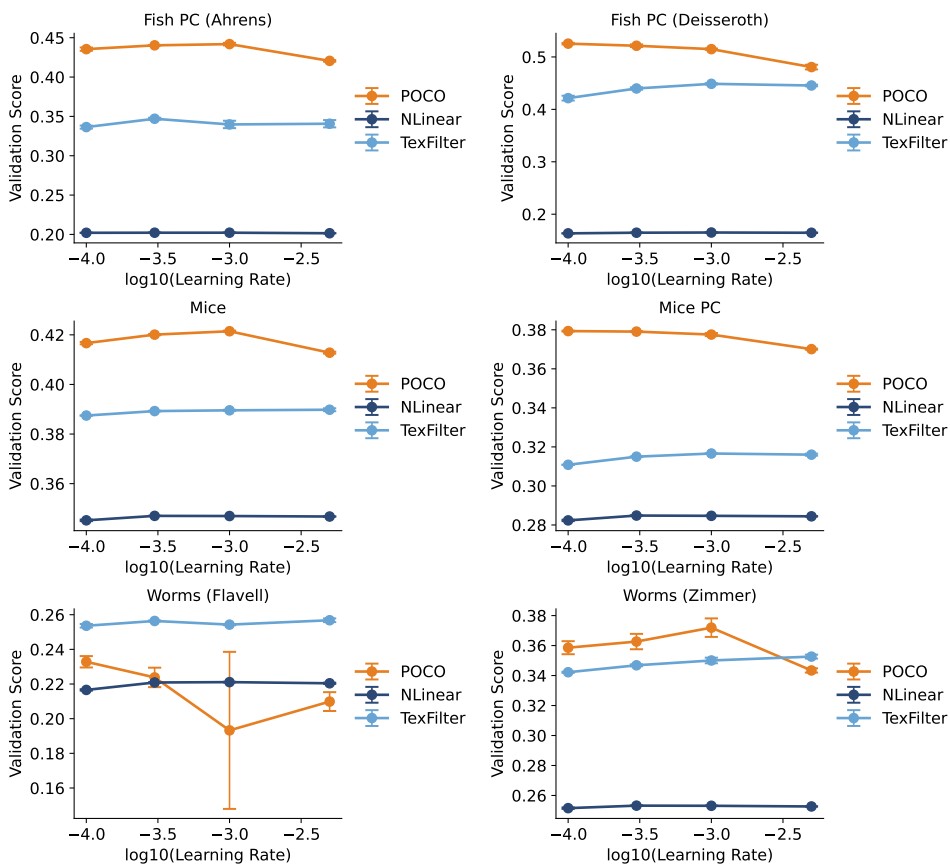

Figure S19: **POCO is robust to a range of learning rates.** Validation scores for different learning rates across datasets. Best performance usually occurs near 0.0003. SEM across 3 seeds.

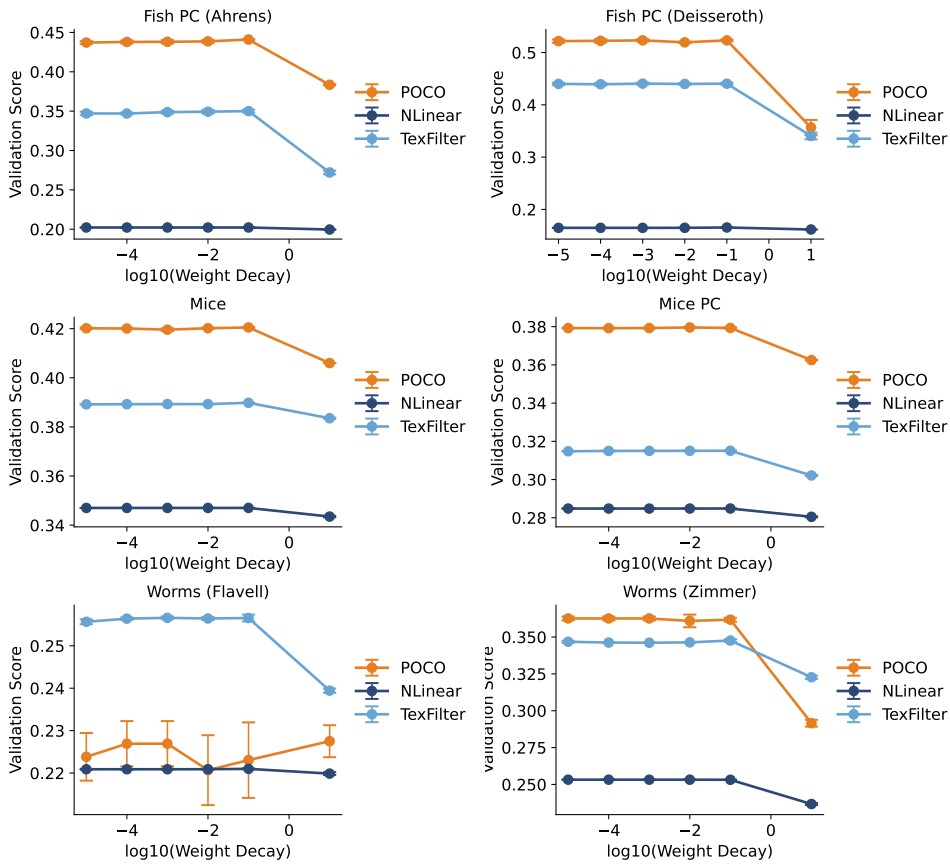

Figure S20: **POCO is relatively insensitive to weight decay settings.** Prediction score across 6 datasets with varied weight decay values. Performance is stable over several orders of magnitude. SEM across 3 seeds.

# B   Datasets

## B.1   Dataset Details

Here we summarize the datasets used in this work. Additional details on the datasets can be found in the original dataset publications.

**Zebrafish (Deisseroth).**   Parts of the dataset are published in [1]. Larval zebrafish expressing nuclear-localized GCaMP6s were used for two-photon calcium imaging. Fish were head-fixed for whole-brain imaging while retaining the ability to perform tail movements. Neural activity was recorded at about 1 Hz, yielding volumetric data from 8,000–22,000 neurons per fish. The imaging data were motion-corrected, segmented into nuclei, and fluorescence traces were extracted and converted into $\Delta F/F$. ROIs were assigned brain region labels based on anatomical landmarks.

Experiments were conducted across four cohorts. The Spontaneous cohort was imaged during natural, unstimulated behavior. The Shocked cohort received randomly timed mild electric shocks during imaging. The Reshocked cohort was pre-exposed to a behavioral challenge before imaging, then imaged under the same shock protocol. The Ketamine cohort received ketamine exposure prior to shock delivery during imaging. There are 5 fish for each cohort, except that the Ketamine cohort has 4 fish. These conditions allowed us to examine brain-wide neural dynamics under varying levels of behavioral and pharmacological perturbation.

**Zebrafish (Ahrens).**  The dataset is published in [11]. A light-sheet imaging setup is used to record whole-brain neural activity from larval zebrafish expressing nuclear-localized GCaMP6f. Imaging was performed at $\sim 2.1$ Hz for $\sim 50$ minutes across 18 fish, resulting in $\sim 6,800$ time points and $\sim 80,000$ segmented neurons per animal. Subject 8, 9, and 11 are excluded due to file incompatibility issues, thus, only 15 fish are used. During imaging, fish were exposed to a battery of visual stimuli, including phototaxis cues, moving gratings (optomotor response), looming discs (escape), dark flashes, and spontaneous illumination blocks. Although behavioral data were also collected, we used only the neural activity traces for modeling, and no stimulus or behavioral labels were provided to the model.

**Mice (Harvey).**   The dataset is partly described in [40], and the mesoscopic calcium imaging technique is detailed in [3]. It includes multi-session two-photon calcium imaging recordings from layer 2/3 neurons in four cortical regions: primary visual cortex (V1), secondary motor cortex (M2), posterior parietal cortex (PPC), and retrosplenial cortex (RSC). Mice expressing GCaMP6s were head-fixed and allowed to run spontaneously on an air-supported spherical treadmill in complete darkness. Imaging was performed using a large field-of-view two-photon microscope, capturing neural activity at about 5.4 Hz from two planes per region. Data were collected across multiple sessions and animals, with each session lasting 45–60 minutes. Four mice were recorded, with 1, 5, 4, and 2 sessions respectively. For the first two mice, all four regions were recorded. For the third mouse, two sessions included simultaneous recordings of PPC, RSC, and V1, while the other two sessions included PPC, RSC, and M2. For the fourth mouse, only PPC, RSC, and V1 were recorded. Motion correction and ROI extraction were performed using standard pipelines. Note that no behavioral or anatomical labels were provided to the model.

*C. elegans* **(Zimmer).** The dataset is described in [23]. Code for data loading and preprocessing is partly based on [25]. This dataset contains whole-brain calcium imaging recordings from *C. elegans*, using a pan-neuronally expressed nuclear calcium indicator GCamp5K. Neural activity was recorded from 5 animals at 2.85 volumes per second for 18 minutes per animal. The imaging volume covered the head ganglia, including most sensory neurons, interneurons, all head motor neurons, and the anterior ventral cord motor neurons. Each recording included 107–131 neurons, with most active neurons identifiable by cell class. Animals were recorded either under constant oxygen or during alternating oxygen stimuli. Only neural activity traces were used in our experiments; stimulus and cell identity labels were not provided to the model.

*C. elegans* **(Flavell).** The dataset is published in [4]. This dataset includes brain-wide calcium imaging recordings from freely moving *C. elegans* using a strain expressing nuclear-localized GCaMP7f and mNeptune2.5 in all neurons. Recordings were conducted on a custom dual-light-path microscope with closed-loop tracking to maintain the worm's head within the imaging field. Neural signals were extracted using automated segmentation and tracking based on 3D U-Net and registration pipelines. We used only the 40 sessions with NeuroPAL neuron identity labels—21 recorded under baseline

conditions and 19 with a noxious heat stimulus that induced a persistent behavioral state change. Only the neural activity traces were used in our experiments; stimulus timing and neuron identity labels were not provided to the model.

**Zapbench.** The benchmark is described in [32]. This dataset consists of whole-brain calcium imaging from a single head-fixed larval zebrafish recorded during fictive behavior in a virtual reality environment. Over the course of a two-hour session, the fish was exposed to nine structured visual stimulus conditions designed to elicit a range of sensorimotor responses. Neural activity was recorded at cellular resolution using light-sheet fluorescence microscopy, resulting in a 4D volumetric dataset spanning 71,721 segmented neurons. Postprocessing included motion correction, custom elastic alignment, and manual neuron segmentation using a Flood-Filling Network. Only the extracted per-neuron calcium traces were used in our experiments; stimulus information was not provided to the model.

### B.2 Data Processing

For Zapbench, we follow the code provided with the benchmark to directly load the processed data [32]. For other datasets, we begin by z-scoring the raw fluorescence values for each neuron so that each calcium trace has zero mean and unit variance. This normalization ensures that all neurons are equally weighted in the model—an active neuron with a high baseline activity contributes similarly to a less active one. For Deisseroth's zebrafish dataset, where raw calcium traces are unavailable, we instead apply z-scoring to the denoised activity extracted using constrained nonnegative matrix factorization (CNMF) [41]. For experiments involving frequency filtering, we apply a zero-phase fourth-order Butterworth filter. The default low-pass filter removes frequency components above $0.1 \times f_s$, where $f_s$ is the sampling frequency. A band-pass filter additionally removes components below $5 \times 10^{-3} \times f_s$. After filtering, we optionally reduce dimensionality using principal component analysis (PCA), applied to the z-scored (and optionally filtered) neural activity matrix. The magnitudes of the resulting principal components are preserved, meaning that dominant components naturally carry more weight in the loss function.

### B.3 Data Partitioning

We first divide each session into segments of 1,000 time steps. Each segment is then split into training, validation, and test partitions using a 3:1:1 ratio. The final segment of a session may contain more than 1,000 steps to maximize use of the available data. To construct the training set, we extract all possible consecutive subsequences of length $C + P$ (64 by default) from the training partition using a sliding window with stride 1. For example, if a session contains 2,500 steps, the training partitions would cover steps $[1, 600]$ and $[1001, 1900]$, and the training subsequences would include $\mathbf{x}_{1:1+C+P}, \mathbf{x}_{2:2+C+P}, \ldots, \mathbf{x}_{600-C-P+1:600}$ and similarly for the second partition. Validation and test sets are constructed in the same way. However, because evaluation is computationally expensive for single-cell resolution zebrafish datasets, we use a larger stride when extracting subsequences from the validation set: stride 32 for Ahrens' dataset and stride 8 for Deisseroth's dataset. Note that striding is applied only to the validation set in the context of single-cell level prediction.

For Zapbench, we instead use the provided tools to load the dataset [32]. One notable difference is that in Zapbench, the dataset is first divided by the stimulus condition then divide each condition into train, validation and test sets.

## C   Additional Model Details

### C.1   POCO

In POCO, we use Rotary Position Embeddings (RoPE) in all attention layers [45]. In RoPE, a timestamp $t$ is assigned to each query token $\mathbf{q}_i$ and key token $\mathbf{k}_j$, and the attention score is computed as

$$\mathbf{a}_{ij} = \mathrm{softmax}\left((\mathbf{R}(t_i)\,\mathbf{q}_i)^T(\mathbf{R}(t_j)\,\mathbf{k}_j)\right), \qquad (8)$$

where $\mathbf{R}(t)$ is a time-dependent rotation matrix. We follow POYO [5] to construct $\mathbf{R}(t)$, defined as a composition of $2 \times 2$ rotation matrices with periods $T_i$ logarithmically spaced between $T_{\min}$ and $T_{\max}$. In our experiments, we use $T_{\min} = 10^{-3} \times T_{\max}$ and set $T_{\max} = 100$ by default.

We also need to define the timestamps. For the first attention layer,

$$\mathbf{L}_1 = \text{Attention}_0(Q = \mathbf{L}_0;\ K, V = \mathbf{E}) \in \mathbb{R}^{N_L \times d}, \qquad (9)$$

we assign timestamps to the trace tokens $\mathbf{E}(i,k)$ as $t_{i,k} = (k-1)T_C$, which corresponds to the starting timestep of the token. For the latent tokens $\mathbf{L}_0(j)$, we set

$$t_j = \frac{j-1}{N_L}C, \quad j \in [N_L], \qquad (10)$$

so that the timestamps uniformly span the input context length $C$. Timestamp for latents in later attention layers $\mathbf{L}_l$ are defined in the same way. For the final layer, the query tokens are given a constant time step $t = C$. Intuitively, the latents can be viewed as encoding the global population state at different moments in the past, and in the final layer, we query how those past states influence the future.

For Zapbench experiments, we modify these parameters to accommodate shorter or longer context lengths: specifically, we use $T_{\max} = 50$, $T_C = 4$ for $C = 4$, and $T_{\max} = 400$, $T_C = 64$ for $C = 256$. The default token length is $T_C = 16$, resulting in 3 tokens per neuron when $C = 48$.

For the conditioning layer, we initialize the weights $\mathbf{W}_\beta, \mathbf{W}_\gamma$ and biases $\mathbf{b}_\beta, \mathbf{b}_\gamma$ of the linear mappings (from the final attention layer output to FiLM conditioning parameters) to zero, ensuring that POCO produces a flat output at initialization. We follow POYO on additional model details: We use an additional feedforward layer following each attention layer. Residual connections are used for both the attention layers and the feedforward layers that follow. We use 1 attention head for the first and last attention layer, and 8 heads for intermediate self-attention layers. We set the embedding and latent size to $d = 128$; in rotary attention, each dimension of each head is 64. We use a dropout of 0.2 in the feed-forward layers following the attention layers and a dropout of 0.4 on the output of intermediate self-attention layers and the accompanying feedforward layers. We initialize the embeddings from $\mathcal{N}(0, \sigma_0^2)$ with $\sigma_0 = 0.02$.

## C.2 Baselines

**NLinear.** In NLinear [55], the last step of the input context is subtracted from the input sequence, and the residual is passed through a linear layer to generate predictions ($\mathbb{R}^C \to \mathbb{R}^P$). The last step is then added back to the output. This design is effective in time series forecasting (TSF) tasks as it helps address distributional shifts between training and test sets. NLinear is a univariate model—each neuron's prediction depends only on its own history—which also makes it naturally capable of processing multi-session data, as it can handle an arbitrary number of neurons.

**DLinear.** We adopt the implementation from [55]. DLinear first decomposes each time series into a trend and a seasonal component. The trend component is obtained by applying a moving average filter with kernel size $2\lfloor C/4 \rfloor + 1$, and the seasonal component is the residual obtained by subtracting the trend from the original input. Both components are predicted independently using linear projections, and the final output is their sum. Like NLinear, DLinear is also univariate.

**MLP.** For single-session experiments, we use a two-layer MLP with ReLU activation and hidden size 1024. This model is equivalent to POCO without FiLM conditioning. For multi-session experiments, we use a 3-layer MLP with hidden size 2048, denoted as MLP_L. MLP is also univariate.

**PLRNN.** We include PLRNN as a baseline due to its demonstrated ability to reconstruct chaotic dynamical systems. We use the implementation from [7]. The model defines a dynamical system over a $d$-dimensional latent state vector $\mathbf{z}_t$ as:

$$\mathbf{z}_t = \mathbf{A}\mathbf{z}_{t-1} + \mathbf{W}\,\phi(\mathbf{z}_{t-1}) + \mathbf{h},$$

where $\mathbf{A} \in \mathbb{R}^{d \times d}$ is a diagonal matrix encoding self-connections, $\mathbf{W} \in \mathbb{R}^{d \times d}$ contains off-diagonal weights for inter-unit interactions, and $\mathbf{h} \in \mathbb{R}^d$ is a bias term. The activation function $\phi$ is ReLU. We set $d = 512$. In our setup, due to the slow dynamics of calcium traces, we use the alternative formulation

$$\mathbf{z}_t = \alpha\,(\mathbf{A}\mathbf{z}_{t-1} + \mathbf{W}\,\phi(\mathbf{z}_{t-1}) + \mathbf{h}) + (1-\alpha)\mathbf{z}_{t-1},$$

where $\alpha = 0.05$. To generate predictions, we first map the last step from the context to the latent state:

$$\mathbf{z}_C = \mathbf{W}_{\text{in}}\mathbf{x}_C + \mathbf{b}_{\text{in}},$$

then evolve the latent state for $P = 16$ steps and map it back to the observation space:

$$\hat{\mathbf{x}}_{C+t} = \mathbf{W}_{\text{in}}^{\dagger}(\mathbf{z}_{C+t} - \mathbf{b}_{\text{in}}), \quad t \in [P],$$

where $\mathbf{W}_{\text{in}}^{\dagger}$ denotes the pseudo-inverse of $\mathbf{W}_{\text{in}}$. PLRNN is multivariate, but it only uses the last time step of the context to generate prediction. To fully utilize the input during training, we instead initialize the latent state $\mathbf{z}_1$ from $\mathbf{x}_1$ and evolve the latent dynamics model for $C + P - 1$ steps. We apply sparse teacher forcing (STF) [35], injecting the ground-truth state every four steps to reduce the drift from the groundtruth for effective training; STF is disabled during the final $P$ prediction steps to simulate free prediction. For multi-session training, the latent dynamics (PLRNN parameters) are shared across sessions, while the input mapping $\mathbf{W}_{\text{in}}$ is session-specific to capture individual variability.

**AR_Transformer.** We also include an autoregressive Transformer [50] as a baseline to assess the performance of classical autoregressive architectures in neural population modeling. A linear layer ($\mathbb{R}^N \to \mathbb{R}^d$) maps each time step's population activity to a $d$-dimensional embedding, producing $C$ tokens. Here we use $d = 512$. These tokens are passed through a 4-layer Transformer with causal attention masks. A final linear projection ($\mathbb{R}^d \to \mathbb{R}^N$) maps the output embeddings to next-step predictions. To forecast $P = 16$ steps, the model is applied autoregressively: at each step, the predicted activity is appended and used as input for the next prediction. As with PLRNN, for multi-session training, the Transformer backbone is shared, while the input and output projection layers are session-specific.

**TSMixer.** TSMixer [16] is an all-MLP architecture for time series forecasting. It consists of stacked mixer blocks, each comprising a time mixer (a linear projection operating along the time dimension) and a feature mixer (an MLP with hidden size 64 that operates across the neuron dimension). The inclusion of the feature mixer makes TSMixer a multivariate model. We use the default hyperparameter settings: two mixer blocks and a dropout rate of 0.1.

**TexFilter.** TexFilter [54] is a recent MLP-based method that incorporates frequency-domain filtering. It applies a context-dependent filter on the Fourier-transformed input, allowing the model to selectively enhance or suppress specific frequency components. In addition to the MLP layer, TexFilter uses learnable complex embeddings that modulate the frequency representation. TexFilter is univariate. In our setup, we set the embedding size to 128 and initialize the embeddings from $\mathcal{N}(0, \sigma_0^2), \sigma_0 = 0.02$, with a dropout rate of 0.3.

**TCN.** ModernTCN [33] employs modernized convolutional blocks for time series modeling. The input is first divided into patches of size $P = 4$, which are embedded into $D = 64$-dimensional vectors. These embeddings are processed using convolutional blocks that capture both temporal and cross-neuron dependencies, followed by a linear readout for prediction. This makes ModernTCN a multivariate model. We follow the original work's hyperparameters for forecasting, except for the convolutional kernel sizes, which are adjusted to 17 (large) and 5 (small) to better suit our shorter context window.

**Netformer.** Netformer [31] is a recent approach for modeling dynamical connectivity in neural population activity. It embeds each neuron's past activity trace and uses an attention layer to compute interaction weights $A$ across neurons. The next-step prediction is given by:

$$\hat{\mathbf{x}}_{t+1} = A\mathbf{x}_t + \mathbf{x}_t.$$

Although the original work only evaluated next-step prediction, we extend it to multi-step forecasting by recursively applying the same update. However, we observed instability when training on long horizons and therefore applied a softmax to the attention weights to stabilize learning. Furthermore, to further improve stability, we also applied an instance normalization layer on the input and later unnormalized the prediction as in recent TSF works [54, 47]. As in the original Netformer, we apply layer normalization along the time dimension, and use an embedding size of 30. Netformer is a multivariate model but is naturally compatible with multi-session training, as the attention mechanism is independent of the number of neurons.

## C.3 Ablation Study

In the ablation study, we considered several alternative architectures. First, to directly use the POYO model to generate 16-steps prediction, we use a linear projection on top of the output of the last layer

$$\mathbf{L}_{L+2} = \text{Attention}_{L+1}(Q = \mathbf{U}_j; K, V = \mathbf{L}_{L+1}) \in \mathbb{R}^{N_j \times d},$$
$$\hat{\mathbf{x}}_{t:t+P} = \mathbf{W}_{out}\mathbf{L}_{L+2}^T + \mathbf{b}_{out}, \tag{11}$$

where $\mathbf{W}_{out} \in \mathbb{R}^{P \times d}$.

Another alternative we tested is replacing the population encoder with a univariate Transformer, which takes the calcium trace of a single neuron instead of encoding the whole population. Specifically, for each neuron $i$, we still use $T_C = 16$ steps to form one token, but the token embedding no longer involves the unit and session embedding:

$$E(k) = \mathbf{W}x^{(j)}_{r_k - T_C:r_k,i} + \mathbf{b}. \tag{12}$$

We then apply a standard 1-layer transformer encoder used in recent time-series forecasting work [47]. Note that the Transformer is applied separately to each neuron. We concatenate hidden states of the last encoder layer, resulting in a $dC/T_C = 128 \times 3$-dimensional vector for each neuron. Finally, we generate 16-step prediction by applying a linear projection on the concatenated hidden state.

Finally, we tested whether POCO's performance could be achieved without the population encoder by relying solely on the unit embedding design. Specifically, we computed the unit embeddings as in POCO, but instead of passing them through the encoder, we directly used linear layers to generate $\boldsymbol{\gamma}$ and $\boldsymbol{\beta}$, which were then used to condition the MLP as in Equation 3. To ensure a fair comparison, we used a larger 3-layer MLP with hidden size 2048 (matching MLP_L), giving it more parameters than POCO, and applied conditioning at the last MLP layer.

# D  Experiments

## D.1  Training

For multi-session training, we train each model for $10^4$ steps. Single-session models are instead trained for $5 \times 10^3$ steps considering the smaller training set size. For all models, we used AdamW [30] optimizer with learning rate 0.0003 and weight decay $10^{-4}$. At each training step, a batch of 64 sequences is sampled from the training sets of all sessions. For single-cell prediction in larval zebrafish, we reduce the batch size to 8 for Deisseroth's zebrafish dataset and to 4 for Ahrens' zebrafish dataset due to memory constraints. Similarly, for the MLP conditioned by univariate Transformer, we reduce the batch size to 16. Gradient clipping is applied to limit the gradient norm to 5. Most models are trained using mean squared error (MSE) loss averaged over the $P$ prediction steps. However, for PLRNN, the autoregressive Transformer, and Netformer—models that generate one-step predictions—the loss is computed across all $C + P - 1$ steps. Models are evaluated on the validation set every 100 training steps, and we report test-set performance using the checkpoint with the lowest validation loss.

## D.2  Simulation

Code for simulation is adapted from [40]. We generate synthetic neural data from a chaotic RNN with a $\tanh$ nonlinearity, governed by the following dynamical equation:

$$\mathbf{r} = \tanh(\mathbf{h}), \quad \tau\frac{d\mathbf{h}}{dt} = -\mathbf{h} + g\mathbf{J}\mathbf{r} + \sqrt{2\tau\sigma^2}\boldsymbol{\xi}, \quad \mathbf{h}(0) \sim \mathcal{U}[-1,1],$$

where $\mathbf{h}$ is the internal (pre-activation) state, $\mathbf{r} = \tanh(\mathbf{h})$ is the firing rate, $\mathbf{J}$ is a recurrent weight matrix, $g = 2.0$ is a gain factor, $\boldsymbol{\xi}$ are $N$ independent Gaussian white noise processes with zero mean and unit variance, $\sigma = 0.1$ controls the noise variance, and $\tau = 0.1$ is the time constant. We discretize the dynamics using Euler's method with time step $\Delta t = 0.01$:

$$\mathbf{h}_{t+1} = \mathbf{h}_t + \frac{\Delta t}{\tau}\left(-\mathbf{h}_t + g\mathbf{J}\tanh(\mathbf{h}_t)\right) + \sqrt{2\sigma^2\frac{\Delta t}{\tau}}\boldsymbol{\xi}_t, \quad \mathbf{r}_t = \tanh(\mathbf{h}_t),$$

where $\boldsymbol{\xi}_t \sim \mathcal{N}(0, I)$ at each step. We z-score the firing rates $\mathbf{r}_t$, and to simulate the slower sampling rate of calcium imaging, we average $\mathbf{r}_t$ over every $f = 5$ time steps. The simulation runs from $t = 0$ to $t = 4096 \times f\tau$, producing 4096 time steps of synthetic data—comparable in length to real neural recordings.

### D.3 Finetuning

For pretraining and finetuning experiments, we partition each dataset and use approximately 80% of the sessions for pretraining. Specifically, for Deisseroth's zebrafish dataset, we reserve the last session from each cohort for finetuning, yielding 15 sessions for pretraining. For Ahrens' zebrafish dataset, we finetune on the last 4 sessions and pretrain on the remaining 11. For the mice dataset from Harvey et al., we finetune using the 2 sessions from the last mouse. All models are finetuned for 2,000 steps and evaluated every 20 steps.

### D.4 Unit Embedding

Although multi-session POCO learns a shared unit embedding space across sessions, the additional session embedding may introduce session-specific offsets. To avoid this confound, we analyze and visualize unit embeddings separately for each session. Let $U(i)$ denote the embedding of neuron $i$, and $S_u^{(j)}$ the set of neurons in region $u$ from session $j$. The cosine similarity between regions $u$ and $v$ is computed as:

$$C(u, v) = \frac{1}{\sum_j |S_u^{(j)}||S_v^{(j)}|} \sum_j \sum_{i_u \in S_u^{(j)}} \sum_{i_v \in S_v^{(j)}} \frac{U(i_u)^T U(i_v)}{\|U(i_u)\|\|U(i_v)\|}, \quad \text{for } u \neq v,$$

$$C(u, u) = \frac{1}{\sum_j |S_u^{(j)}|(|S_u^{(j)}| - 1)} \sum_j \sum_{\substack{i_u, i_v \in S_u^{(j)} \\ i_u \neq i_v}} \frac{U(i_u)^T U(i_v)}{\|U(i_u)\|\|U(i_v)\|}.$$

To highlight relative similarity patterns, we compute a row-wise normalized similarity:

$$\bar{C}(u, v) = \frac{C(u, v) - \min_{v'} C(u, v')}{\max_{v'} C(u, v') - \min_{v'} C(u, v')}.$$

This normalized similarity more clearly reveals which brain regions are functionally closer or further from a given region $u$. The same analysis can be applied using other distance metrics, such as Euclidean distance.

### D.5 Multi-Dataset Training

To support multi-dataset training, we assign a separate dataloader to each dataset. At each training step, we sample one batch from each dataset, compute the loss independently, and sum the losses before performing a single backward pass. To help POCO handle cross-dataset differences, we add a dataset-specific embedding to each input token:

$$E(i, k) = \mathbf{W} x_{r_k - T_C : r_k, i}^{(j)} + \mathbf{b} + \text{UnitEmbed}(i, j, u) + \text{SessionEmbed}(j, u) + \text{DatasetEmbed}(u),$$

where $u \in [D]$ is the dataset index and $D$ is the total number of datasets.

### D.6 Experiments on Zapbench

For Zapbench, we follow the provided script for dataloading and testing [32]. We train POCO for 25 epochs with a batch size of 8, validating after every epoch. We report test-set performance using the checkpoint with the lowest validation loss. Following the benchmark protocol, we use mean absolute error (MAE) as the loss function. The *TAXIS* condition is excluded during training, and performance is evaluated on the remaining stimulus conditions.

## E   Additional Discussion

While POCO generally outperforms baseline models, its gains are smaller in certain settings. We believe these variations reflect differences in dataset properties such as species, signal-to-noise ratio

(SNR), recording duration, and population size. Datasets with lower SNR impose an effective noise ceiling on performance, reducing the apparent gap between models since no method can explain the irreducible variance. In such cases, simpler models like MLPs may appear relatively stronger as they are less prone to overfitting.

For example, the C. elegans datasets involve small populations and slow, graded dynamics, which likely limit the benefits of population-level conditioning. The performance drop after low-pass filtering may occur because C. elegans activity already evolves slowly—further filtering can remove subtle but informative temporal variations. Similarly, the smaller gain of POCO in single-neuron zebrafish data compared to PCA-reduced activity likely reflects higher noise levels and sparser activations at the single-cell level.

Another possible factor is the calcium indicator itself, as fluorescence dynamics differ across indicators (e.g., GCaMP6s, 6f, 7f). In the current implementation, we deliberately chose not to explicitly model calcium kinetics or deconvolved spikes, aiming to maintain flexibility across datasets with different indicators and preprocessing pipelines. Future work could explore incorporating indicator-specific dynamics or explicit deconvolution to improve generalization and interpretability across modalities.

More broadly, there are theoretical limits to predictability in neural forecasting. Even with large datasets and expressive models, many factors—such as stochastic spiking, unobserved inputs, and measurement noise—introduce intrinsic uncertainty in future activity. Under an MSE objective, models tend to output the average of possible futures, which can flatten predictions in regions of high uncertainty. Modeling uncertainty explicitly, for instance via probabilistic or diffusion-based forecasting approaches, could help capture this variability.

Finally, we note that in the time-series forecasting literature, new architectures rarely achieve top performance across all datasets [33, 16, 54], suggesting that variability in model ranking is a broader characteristic of modeling complex, noisy systems rather than a limitation unique to POCO. We hope that the remaining performance gaps observed here will motivate future research on how dataset properties and noise structures influence the limits of neural forecasting.

## F  Compute Resources

All models are trained on NVIDIA A100 or H100 GPUs paired with AMD EPYC 9454 CPUs. Only a single CPU thread is used throughout training. Data preprocessing for all datasets takes less than one CPU day in total, and generating all synthetic data requires under two CPU days.

For multi-session prediction of the first 512 principal components (PCs) from Deisseroth's zebrafish dataset, training POCO for $10^4$ steps takes less than 30 minutes, including data loading and validation time. Despite using a high-end GPU, this training setup consumes under 4GB of GPU memory. Computational cost increases for datasets with more neurons: for single-cell prediction on Ahrens' zebrafish dataset, training consumes up to 48GB of GPU memory and completes in under one hour.

While training a single model is fast, some experiments involve training many models. For example, training a single-session model for each session in each dataset across 4 random seeds involves training over 300 models in total. The estimated total GPU time across all experiments is approximately 1,500 hours.

The same hardware setup (H100 GPU) is used to measure finetuning and inference speed. When finetuning POCO on the first 512 PCs of Deisseroth's dataset, we find that adapting the embedding for 200 training steps takes less than 15 seconds, and forecasting a single sequence takes only 3.5 milliseconds.

