# OpenReview forum: "POCO: Scalable Neural Forecasting through Population Conditioning"
_NeurIPS.cc/2025/Conference — NeurIPS 2025 poster_

### Official Review · Reviewer_KEzN · 2025-06-16

**Clarity:** 4
**Significance:** 2
**Originality:** 2
**Rating:** 4
**Confidence:** 3

**Summary:**

The authors propose a new model POCO for forecasting neural activity (specifically from calcium imaging), building on POYO and FiLM. They demonstrate its SOTA performance in single- and multi-session settings across several organisms (zebrafish, mice, C. elegans). They further perform analyses on effects of hyperparameters and architectural choices to justify and inform their model design.

**Questions:**

1. As noted above, POCO gains over MLP don't seem that substantial outside of the two zebrafish PC forecasting tasks. Can simply scaling the MLP size further (to e.g., match parameter count of full POCO) make the simple MLP match or outperform POCO? Do you have any explanation for why the conditioning helps more in some cases than others?
2. What hyperparameter tuning if any was done for TSMixer (and other baselines)? It appears the architecture described in the Appendix is smaller (hidden dimension of 64) compared to the optimal architecture found by Lueckmann et al. for their ZAPBench baselines.
3. Do you have any explanation for the much poorer performance of POCO on C. elegans data after low-pass filtering?
4. Are the reported MLP results for the finetuning evaluation (Figure S12) MLPs trained from scratch? If yes, can you report these curves if you used pre-trained multi-session MLPs?

This is a bit much to ask in the rebuttal period, but if there is time I would appreciate seeing this addressed too:

5. Is it possible to add a new baseline that is just the MLP but with learnable unit embeddings as an additional input (either concatenated with the neural activity or used as modulation in FiLM)? I would be especially curious to see this in the pre-training -> fine-tuning unit embedding evaluations. Along with this, I would be curious to see the zero-shot and finetuned performance of the transformer-conditioned MLP from the ablation study. For me, these extra baselines could strengthen the argument for the use of the POYO encoder over something that simpler/more parallelizable does not consider the rest of the population.

**Ethical Concerns:**

["NO or VERY MINOR ethics concerns only"]

**Final Justification:**

The additional experiments described in the rebuttal strengthen the authors' case that the complexities of POCO do lead to a fairly reliable performance benefit. Still, POCO in many cases doesn't seem to be that much better than just an MLP, and some of the variance in performance across datasets is a bit concerning. However, I really appreciate the overall thoroughness of the work and think the extensive evaluations in the study are worth sharing with the community and can lead to further work that may give us better understanding of when population encoding is beneficial to forecasting and what architectures are better suited to different datasets. For this reason, I recommend acceptance.

**Limitations:**

yes

**Quality:**

3

**Strengths And Weaknesses:**

**Strengths:**
* Text is clearly written and quite easy to understand.
* The architecture itself is simple and reasonable, building on two established architectures (POYO encoder and FiLM).
* The ablations and analyses of how performance varies with multi-session/subject training, dataset size, and architecture and hyperparameter settings are thorough and much appreciated.

**Weaknesses:**
* I worry that baselines were not tuned enough, though I understand that is a huge burden for the authors. But, in particular, the poor performance of TSMixer (worse than the copy baseline in many cases) is a bit strange when it performs well in ZAPBench. In fact, it substantially outperforms POCO in the long context setting, a fact I think should be more clearly stated in the text---it currently reads as if only the UNet bests POCO.
* POCO does not seem to substantially outperform the simple MLP except for the two zebrafish datasets in the PC forecasting setting. In addition, when the C. elegans data are low-pass filtered, POCO performance drops sharply and significantly underperforms MLP, which raises doubts about POCO's robustness.
* There is (to my understanding) no comparison in the finetuning evaluation to a pre-trained MLP, only to an MLP trained from scratch, but I think POCO needs to demonstrably outperform a pre-trained univariate MLP to prove useful in this setting.
* The architecture draws heavily from prior work, and the results on structure in unit embeddings, multi-session/subject training, pre-training benefits, etc. seem generally consistent with prior work as well (POYO, NDT1-3). So, while the application to calcium trace forecasting is certainly new, the overall novelty of those results is a bit limited.

In summary, I appreciate the extensive evaluations and analyses in the paper, but without some further details/evaluations, I remain somewhat unconvinced of POCO's utility over even just a univariate MLP, both for conventional forecasting setting as well as the pre-training/finetuning setting.

---

> ### Author Rebuttal · Authors · 2025-07-30
>
> Thanks for your careful review and constructive feedback!
>
> > Can simply scaling the MLP size further (to e.g., match parameter count of full POCO) make the simple MLP match or outperform POCO?
>
> Thank you for the question. In the current manuscript, the MLP baseline is identical to the MLP component used within POCO (2 layers, hidden size = 1024), which allows us to isolate the effect of population conditioning. However, we agree that comparing against larger MLPs with parameter counts similar to POCO is informative, so we evaluated deeper and wider MLPs:
>
> |Model (MLP layers,MLP hidden size)|Parameter Count|Fish PC(Deisseroth)|Fish PC(Ahrens)|Mice|
> |---|---|---|---|---|
> |MLP(2,1024)|0.07M|0.418±0.003|0.370±0.002|0.409±0.000|
> |MLP(2,2048)|0.14M|0.421±0.002|0.370±0.001|0.409±0.000|
> |MLP(3,2048)|4.3M|0.466±0.001|0.418±0.001|0.412±0.000|
> |MLP(4,2048)|8.5M|0.471±0.002|0.417±0.006|0.412±0.000|
> |POCO(2,1024)|2.6M–4.0M|**0.525±0.004**|0.440±0.003|**0.420±0.002**|
> |POCO(4,1024)|4.6M–6.1M|**0.525±0.012**|**0.449±0.001**|**0.420±0.001**|
>
> We find that increasing MLP depth does improve its performance, particularly on the zebrafish datasets. However, POCO still consistently outperforms MLPs with comparable or larger parameter counts, indicating that model size alone does not explain the performance gap. Based on these findings, we plan to include a larger MLP baseline (3 layers, hidden size = 2048) in the final paper and include the table in appendix. We also acknowledge that further exploration may even yield stronger baselines. However, the same architectural enhancements could also be applied to POCO’s MLP forecaster, likely improving its performance as well. We view this as a valuable direction for future work.
>
> > Is it possible to add a new baseline that is just the MLP but with learnable unit embeddings as an additional input (either concatenated with the neural activity or used as modulation in FiLM)?
>
> Thank you for the suggestion. We agree that including these additional baselines strengthens the evaluation and more rigorously demonstrates the advantage of POCO. As you proposed, we tested a model where an MLP is directly conditioned on unit embeddings using FiLM ("UMLP"). We also added session embedding so the embedding scheme is exactly the same as POCO. We used the large MLP in earlier experiments (3 layers, hidden size = 2048).
>
> | Model | Fish PC (Deisseroth) | Fish PC (Ahrens) | Mice |
> | --- | --- | --- | --- |
> | POCO | **0.525 ± 0.004** | **0.440 ± 0.003** | **0.420 ± 0.002** |
> | MLP (large) | 0.466 ± 0.001 | 0.418 ± 0.001 | 0.412 ± 0.000 |
> | UMLP (large) | 0.481 ± 0.004 | 0.434 ± 0.004 | 0.413 ± 0.000 |
>
> We find that adding unit embeddings improves the MLP’s performance slightly, particularly when combined with a larger network. However, POCO still outperforms these variants, suggesting that the unit embedding can only account for part of the performance gain of POCO. We will add these results to the ablation study section in the updated manuscript.
>
> > Are the reported MLP results trained from scratch? If yes, can you report these curves if you used pre-trained multi-session MLPs?
> > Along with this, I would be curious to see the zero-shot and finetuned performance of the transformer-conditioned MLP from the ablation study. For me, these extra baselines could strengthen the argument for the use of the POYO encoder over something that simpler/more parallelizable does not consider the rest of the population
>
> Thank you for pointing this out. Indeed, the MLP models in Figure S12 were trained from scratch. We agree that evaluating fine-tuned versions of MLP and the new baselines would further clarify POCO’s utility in this setting. As uploading new figures is not permitted, we summarize the final fine-tuned results below as Table S9. We also include zero-shot performance for transformer-conditioned MLPs (TMLP) and standard MLPs.
>
> |Model| Fish PC (Deisseroth) | Fish PC (Ahrens) | Mice |
> |---|---|---|---|
> |Pre-POCO(full finetune)|**0.532±0.014**|**0.424±0.017**|**0.406±0.001**|
> |Pre-POCO(embedding only)|**0.539±0.008**|**0.392±0.062**|**0.405±0.001**|
> |Pre-MLP|0.446±0.002|0.385±0.003|0.398±0.000|
> |Pre-TMLP|0.470±0.002|**0.423±0.017**|0.400±0.000|
> |Pre-MLP(large)|0.470±0.006|**0.432±0.007**|0.401±0.000|
> |Pre-UMLP(large,full finetune)|0.495±0.003|**0.427±0.013**|0.404±0.000|
> |Pre-UMLP(large,embedding only)|0.499±0.001|0.400±0.007|0.404±0.000|
> |Pre-TMLP(zero-shot)|0.471±0.001|0.362±0.007|0.397±0.000|
> |Pre-MLP(large,zero-shot)|0.474±0.001|0.369±0.002|0.399±0.001|
>
> Overall, pre-trained POCO achieves the best performance across datasets, although some models perform similarly on Ahrens’ dataset. We will update these results in the paper.
>
> While the performance gains may appear modest, we emphasize that predictive accuracy is not the only motivation for using POCO over a simple univariate forecaster. A major goal of neural prediction is to obtain models that can support downstream tasks such as studying effective connectivity [1] or enabling closed-loop optogenetic control [2]. Univariate forecasters lack population-level modeling and are not applicable to these use cases.
>
> [1] Luo, Zixiang, et al. "Mapping effective connectivity by virtually perturbing a surrogate brain." *Nature Methods*
> [2] Grosenick, Logan, James H. Marshel, and Karl Deisseroth. "Closed-loop and activity-guided optogenetic control." *Neuron*
>
> > I worry that baselines were not tuned enough, though I understand that is a huge burden for the authors… TSMixer substantially outperforms POCO in the long context setting, a fact I think should be more clearly stated in the text… What hyperparameter tuning if any was done for TSMixer (and other baselines)? It appears the architecture described in the Appendix is smaller (hidden dimension of 64) compared to the optimal architecture found by Lueckmann et al. for their ZAPBench baselines.
>
> Thank you for pointing this out. We will revise the text to more clearly acknowledge POCO’s limitations on ZAPBench. However, we note that ZAPBench includes only a single session, and POCO’s performance can vary across sessions even within the same dataset.
>
> As described in the Appendix, we started from default hyperparameters in the original papers and made necessary adaptations to ensure baselines ran properly on our datasets (particularly for ModernTCN, NetFormer, and PLRNN). In Figures S18 and S19, we explored learning rate and weight decay effects and found model performance to be generally robust to these choices.
>
> Given the large hyperparameter space and the number of datasets, sessions, and baselines, we did not extensively tune all parameters—especially model size. This is particularly challenging for single-session models like TSMixer, which require training a separate model per session, increasing computational cost.
>
> That said, we agree model size can be sensitive. To address this, we ran additional TSMixer experiments and found that larger models tended to overfit and perform worse:
>
> |Model|Fish PC(Deisseroth)|Fish PC(Ahrens)|Mice|
> |---|---|---|---|
> |TSMixer 64|**-0.546±0.066**|**0.013±0.009**|**0.391±0.001**|
> |TSMixer 128|-0.688±0.014|-0.017±0.010|0.384±0.003|
> |TSMixer 256|-0.788±0.050|-0.031±0.012|0.376±0.001|
>
> We also tested larger model sizes for PLRNN, AR_Transformer, NetFormer, and TexFilter, and did not observe substantial performance improvements.
>
> > POCO does not seem to substantially outperform the simple MLP except for the two zebrafish datasets in the PC forecasting setting. … when the C. elegans data are low-pass filtered, POCO performance drops sharply
>
> Thank you for the comment. We agree that POCO’s gains appear smaller in some settings, which is likely due to differences in signal quality and dataset characteristics. Unpredictable noise in the recordings effectively imposes a noise ceiling on performance, pulling the prediction scores of different models closer together, as no model can explain the irreducible variance. We also find that POCO’s advantage is more pronounced in PCA-reduced space, where uncorrelated noise is suppressed. In contrast, single-neuron activity is sparser and noisier, and performance is averaged across neurons, which reduces measured gains.
>
> Regarding the performance drop after low-pass filtering on *C. elegans*, we believe this is because *C. elegans* neurons already exhibit slow, sparse, and graded dynamics as opposed to spike trains. Further filtering may remove subtle but informative temporal variations, making POCO easier to overfit to the training set, and thus flattening out any advantage over simpler models.
>
> Due to space constraints, please see the second part of our response to the second Reviewer (xBNB) for further discussion on why we believe this does not diminish the contribution of our work.
>
> > The architecture draws heavily from prior work, and the results seem generally consistent with prior work … while the application to calcium trace forecasting is certainly new, the overall novelty of those results is a bit limited.
>
> While POCO builds on existing components (e.g., MLPs, FiLM, Perceiver-based encoders), its core architectural idea—modulating univariate forecasters with population-level context for multi-step calcium prediction—is, to our knowledge, novel and specifically designed for the challenges of neural forecasting across sessions.
>
> We respectfully note that the method itself is only part of our contribution. We evaluate all models on five diverse datasets spanning three species and nearly 50 hours of recording. To our knowledge, the multi-species training setup and accompanying simulation are also novel. Moreover, the scale of datasets and breadth of baselines far exceeds recent benchmarks like ZapBench. We believe this comprehensive evaluation—along with analyses of generalization, adaptation, and interpretability—adds significant value to the field.

---

> > ### Comment · Reviewer_KEzN · 2025-08-02
> >
> > Thank you for the response! I really appreciate the new experiments and they convinced me that the population encoding can provide an actual benefit. I am still somewhat concerned about the low-pass filtered C. elegans result and am curious what solutions to overfitting, if any, were tried to fix it. Regardless, I think the overall thoroughness of the work is informative and worth sharing so I will raise my score.

---

> > > ### Author Response · Authors · 2025-08-04
> > >
> > > Thank you for your kind follow-up and for reconsidering your evaluation. We're glad the new experiments helped clarify the benefits of population encoding. Regarding the low-pass filtered *C. elegans* result, we focused our model development primarily on non-filtered datasets and did not tune hyperparameters separately for each dataset.
> > >
> > > That said, we’ve since conducted additional experiments and found that reducing model size, lowering the learning rate, and adding an extra MLP layer can alleviate overfitting in this setting:
> > >
> > > | Model (Learning Rate, MLP Layers, Embedding Size) | Worm (Flavell, low-pass) | Worm (Zimmer, low-pass) |
> > > | --- | --- | --- |
> > > | POCO (0.0003, 2, 128) — Default | 0.088 ± 0.006 | 0.314 ± 0.027 |
> > > | POCO (0.0001, 2, 16) | 0.150 ± 0.146 | **0.380 ± 0.029** |
> > > | POCO (0.0001, 2, 128) | 0.155 ± 0.338 | 0.325 ± 0.032 |
> > > | POCO (0.0001, 3, 16) | **0.275 ± 0.101** | **0.379 ± 0.041** |
> > > | POCO (0.0001, 3, 128) | 0.035 ± 0.073 | 0.325 ± 0.018 |
> > > | TexFilter | *0.423 ± 0.003* | *0.407 ± 0.003* |
> > >
> > > While TexFilter still achieves the best overall performance, the gap narrows considerably when POCO is adjusted to a smaller, more regularized configuration. We plan to include these results in the supplementary material of the camera-ready version.

---

### Official Review · Reviewer_pkKL · 2025-06-18

**Clarity:** 4
**Significance:** 2
**Originality:** 2
**Rating:** 4
**Confidence:** 3

**Summary:**

This paper proposes an efficient new model, POCO, for predicting future neural activity, by combining single-neuron forecasting with a light-weight model of interaction. This emphasizes applications to longer recordings of freely behaving animals. POCO is evaluated experimental on five calcium imaging datasets of free behavior in three different species where it outperforms all baselines, generalizes to new recordings, and recovers physiologically-relevant structure in its representations.

**Questions:**

Q1. The sampling rate varies by a factor of 5 across the different datasets, which using a 16 step prediction window would imply that only about 2ms of activity need to be predicted for mouse versus 10ms of activity of zebrafish. Is this being controlled for somehow? If not, why is it unimportant?

Q2. I understand that given resource limitations it's reasonable to do much of the model's evaluations on relatively small datasets. However, it would be very helpful to see a few evaluations on a significantly longer dataset given this is an important part of the gap the paper intends to address.

Q3. Did you compare against models like those of Azabou et al. [1] or Zhang et al. [2]? I am willing to raise my score if there is a convincing case that these models fall short in the paradigm considered here.

Q4. Is the same-sized model used in the multi-species training setup? Conceivably, the larger aggregate dataset could support a larger model.

[1] Azabou et al. "Multi-task, multi-session neural decoding..." ICLR 2025
[2] Zhang et al. "Towards a 'universal translator'..." NeurIPS 2024

**Ethical Concerns:**

["NO or VERY MINOR ethics concerns only"]

**Final Justification:**

The core problem addressed in this paper -- forecasting in low-frequency calcium imaging datasets -- is an understudied one, if not particularly interesting, and this paper contributes both an effective model using well-established ideas and a comprehensive evaluation of different methods for this problem. The evaluation is especially rigorous and thorough and in my view is this paper's main contribution.

**Limitations:**

Yes

**Quality:**

3

**Strengths And Weaknesses:**

Strengths

S1. Foundation models, generative ones in particular, are a fascinating new direction for neuroscience that may dramatically alter the field, and basic research on them is important and timely.

S2. The experimental evaluation is comprehensive, thoughtful, and carefully explained.

S3. The paper is well-written and clear throughout.

Weaknesses

W1. The gap in the related work is not entirely clear to me. The motivation given for the particular model examined in this paper emphasizes longer freely behaving recordings in contrast to short task-focused ones, but is this actually important? Do foundation models trained on the one really fail to generalize to the other? It's plausible, but it's an empirical question that needs investigation. Notably, those foundation models are not compared against in this paper, and the datasets examined here represent only a few seconds of neural activity -- far from long periods of free behavior. Also, why do we need a different architecture for these paradigms?

W2. It would be nice to have a demonstration, even a small one, where forecasting is useful. In general predicting neural activity is an interesting problem but a solution to it should either develop our understanding of the brain or demonstrably improve the performance of an applied neuro technology.

W3. The proposed method is a very basic extension of existing ideas and models which are already being used for closely related tasks.

---

> ### Author Rebuttal · Authors · 2025-07-30
>
> Thanks for your careful and constructive review!
>
> > the datasets examined here represent only a few seconds of neural activity -- far from long periods of free behavior.
>
> First, we would like to clarify the timescale of the datasets used in this work. In Table 1, the sampling frequency $f\_s$ denotes the number of time steps (i.e., imaging frames) per second. Each session thus contains tens of minutes of recording, and 16-step prediction in the Deisseroth zebrafish dataset (with $f\_s = 1.1$ Hz) corresponds to forecasting approximately 15 seconds into the future.
>
> > It would be very helpful to see a few evaluations on a significantly longer dataset given this is an important part of the gap the paper intends to address.
>
> We hope the clarification above addresses this concern. In total, the datasets used in our study comprise nearly 50 hours of recordings, which we believe provide a substantial basis for testing and benchmarking forecasting models. That said, as noted in the Discussion, scaling and refining our approach for even larger datasets remains an important direction for future work.
>
> > The sampling rate varies by a factor of 5 across the different datasets, which using a 16 step prediction window would imply that only about 2ms of activity need to be predicted for mouse versus 10ms of activity of zebrafish. Is this being controlled for somehow? If not, why is it unimportant?
>
> As you correctly pointed out, the sampling rates across datasets differ significantly. We chose not to downsample the data to a uniform frequency in order to make maximal use of the available data. In Figure 2A, we show that POCO outperforms baseline models across all 16 future steps, suggesting that the specific prediction length is not the primary factor determining relative model performance.
>
> However, we agree that downsampling to a consistent rate could offer useful insight. To investigate this, we downsampled Ahrens’ zebrafish dataset by a factor of 2 and the mouse dataset by a factor of 5, so that the effective sampling rate for all three datasets is approximately 1 Hz, matching Deisseroth’s zebrafish dataset.
>
> |Model|Fish PC(Deisseroth)|Fish PC(Ahrens)|Mice|
> |---|---|---|---|
> |POCO|**0.525±0.004**|**0.452±0.005**|0.455±0.001|
> |MLP|0.417±0.002|0.398±0.002|**0.459±0.000**|
> |NLinear|0.165±0.000|0.265±0.001|0.415±0.000|
> |TexFilter|0.440±0.005|0.384±0.006|0.434±0.000|
> |NetFormer|0.221±0.008|0.264±0.005|0.416±0.000|
>
> We found that overall model performance remains consistent, though POCO performs slightly worse than MLP on the mouse dataset—likely because downsampling by a factor of 5 reduces the effective dataset size. This aligns with our findings in Figure 3A, where simpler models like MLP tend to perform better in low-data regimes.
>
> > The gap in the related work is not entirely clear to me. The motivation given for the particular model examined in this paper emphasizes longer freely behaving recordings in contrast to short task-focused ones, but is this actually important? … Also, why do we need a different architecture for these paradigms?
> > Did you compare against models like those of Azabou et al. [1] or Zhang et al. [2]?
>
> Thank you for pointing out this potentially confusing aspect of our motivation. There are two key differences between our work and prior studies on neural foundation models:
> (1) We focus on forward prediction of calcium imaging data, whereas most previous work has focused on spiking activity and/or behavioral decoding.
> (2) Our datasets consist primarily of long recordings in freely behaving animals, while prior work often uses task-driven settings.
>
> Importantly, the need for a different model architecture in our case is not due to (2)—as you correctly noted, models can often be applied to both task-focused and freely behaving datasets. Rather, the main challenge stems from (1): models developed for spiking data or behavioral decoding are not directly applicable to the problem of predicting high-dimensional calcium traces.
>
> You mentioned two recent works [1, 2], but it is not straightforward to adapt a behavioral decoding model [1] for forward forecasting, nor to apply a model designed for spiking data [2] to calcium imaging. Specifically, [2] trains on data with high temporal precision (20 ms bins) and predicts ~200 ms, whereas calcium imaging has much lower temporal resolution. Additionally, their spatial chunking strategy was designed for ~600 neurons per session for spiking data. Scaling this to tens of thousands of calcium traces across sessions is nontrivial – the time complexity of methods [2] grows quadratically with the number of neurons. The spatial chunking strategy is also hard to interpret if we aim to model PCs. In contrast, our task is more naturally aligned with time-series forecasting, which motivates our comparisons to general-purpose forecasting models.
>
> That said, we agree that adapting more models for comparison is worthwhile, and we have already included an adaptation of [1] in our ablation study. Specifically, we tested a POYO-only variant where predictions are generated by querying the latent embedding directly, instead of using FiLM and MLP (Table 5). We found that this version produces predictions that are often disconnected from the context, which we believe is because the model cannot retain detailed information about current activity in each channel. Its performance does not consistently outperform the copy baseline, suggesting that while POYO may be effective for modeling global brain state or low-dimensional behavioral variables, it struggles with the fine-grained dynamics of large neural populations.
>
> [1] Azabou et al. "Multi-session, multi-task neural decoding from distinct cell-types and brain regions
> " ICLR 2025 [2] Zhang et al. "Towards a “universal translator” for neural dynamics at single-cell, single-spike resolution" NeurIPS 2024
>
> > Is the same-sized model used in the multi-species training setup?
>
> In the paper, we reported results using a model of fixed size. However, we also tested larger POCO variants in the multi-species setting and did not observe significant performance improvements:
>
> |Model(layers,hidden size)|Fish PC(Deisseroth)|Fish PC(Ahrens)|Mice|Worms(Zimmer)|Worms(Flavell)|
> |---|---|---|---|---|---|
> |POCO(1,128)|0.500±0.001|0.441±0.007|0.402±0.002|0.324±0.010|0.250±0.011|
> |POCO(2,256)|0.497±0.009|0.440±0.002|0.402±0.003|0.318±0.020|0.252±0.019|
> |POCO(4,512)|0.494±0.009|0.442±0.007|0.401±0.002|0.321±0.008|0.250±0.025|
>
> These results further support our view that multi-species forecasting challenges arise from biological and dataset heterogeneity, rather than insufficient model capacity.
>
> > It would be nice to have a demonstration, even a small one, where forecasting is useful. In general predicting neural activity is an interesting problem but a solution to it should either develop our understanding of the brain or demonstrably improve the performance of an applied neuro technology.
>
> We agree that showcasing additional use cases could further strengthen the paper. However, we also believe that neural prediction is a sufficiently fundamental problem to be studied on its own (e.g., consider the line of work following Brain-Score [1] in vision neuroscience or the recent Zapbench [2]). Prior studies have shown that predictive models can be used to infer effective connectivity [3], and our analysis of unit embeddings suggests that POCO may also help interpret the contributions of individual neurons to population dynamics.
>
> Forward models of neural activity are also critical for closed-loop or activity-guided optogenetic control, as they can be used to simulate the effects of stimulation [4]. They are particularly valuable in systems aiming to suppress seizures, where accurate forecasting can enable early intervention [5]. However, implementing such systems requires advanced imaging and hardware infrastructure, which is beyond the scope of this work.
>
> [1] Schrimpf, Martin, et al. "Brain-score: Which artificial neural network for object recognition is most brain-like?." [2] Lueckmann, Jan-Matthis, et al. "ZAPBench: A Benchmark for Whole-Brain Activity Prediction in Zebrafish. [3] Luo, Zixiang, et al. "Mapping effective connectivity by virtually perturbing a surrogate brain." Nature Methods [4] Grosenick, Logan, James H. Marshel, and Karl Deisseroth. "Closed-loop and activity-guided optogenetic control." Neuron [5] Kang, Wonok, et al. "Closed-loop direct control of seizure focus in a rodent model of temporal lobe epilepsy via localized electric fields applied sequentially." Nature Communications
>
> > The proposed method is a very basic extension of existing ideas and models which are already being used for closely related tasks.
>
> While POCO builds on existing components (e.g., MLPs, FiLM, and Perceiver-based encoders), the core architectural idea—using population-level context to modulate per-neuron univariate forecasters for multi-step calcium activity prediction—is, to our knowledge, novel and tailored to the unique challenges of neural forecasting across sessions.
>
> We also respectfully note that the proposed method is only one part of our contribution. The paper systematically explores neural prediction by using models from multiple research fields. Importantly, we evaluate all models on five diverse datasets spanning three species, including nearly 50 hours of recording in total. The scale of the datasets as well as the number of models tested are much larger than recently published benchmarking work on neural prediction [2]. We believe this breadth of experimentation, along with detailed analyses of generalization, adaptation, and interpretability, constitutes a valuable contribution to this emerging field which lacks controlled comparisons of different methods.

---

> > ### Comment · Reviewer_pkKL · 2025-08-03
> >
> > Thank you for the thoughtful response, and especially for correcting my abject misreading of Table 1. At this point, I agree that there is a worthwhile contribution within this paper -- but it's suppressed by the current framing in the abstract, introduction, and discussion, which have nothing to say about (i) the particular difficulties of modeling calcium imaging datasets or (ii) benchmarking forecasting performance on large, diverse datasets, outside of a brief mention on lines 50-52. I'm raising my score to a 4, but I do strongly encourage the authors to revised the framing of the paper for the camera-ready version to address these points.

---

> > > ### Author Response · Authors · 2025-08-04
> > >
> > > Thank you for your thoughtful follow-up and for reconsidering your evaluation. We appreciate your candid feedback and are glad the clarification was helpful. We fully agree that the contribution of the paper can be more clearly highlighted by discussing the unique challenges of modeling calcium imaging data and emphasizing the value of benchmarking forecasting performance across large, diverse datasets. We will revise the framing accordingly in the camera-ready version to better communicate these points.

---

### Official Review · Reviewer_xBNB · 2025-06-27

**Clarity:** 3
**Significance:** 2
**Originality:** 3
**Rating:** 4
**Confidence:** 4

**Summary:**

This paper presents a novel neural forecasting framework that integrates a univariate forecaster with a population encoder, trained jointly across multiple sessions of *spontaneous calcium imaging* data. The authors demonstrate that it achieves state-of-the-art predictive accuracy compared to existing baselines for multiple species.

**Questions:**

- Prediction Visualization: To better understand the differences among the models, it would be useful to include a visual comparison of the predictions from POCO, MLP, and TexFilter—either by extending Fig. 2C or by adding a supplementary figure. This could help illustrate the qualitative distinctions that are not fully captured by the prediction metrics.
- Context Length Saturation: I am also puzzled by the observation that POCO’s performance saturates rapidly with increasing context length. Could the authors provide some insight into why this might be happening?
- Learnable Latents ($L_0$): Could the authors clarify the learnable latent variables $L_0$? Are these latents shared across all time slices and sessions, or are they session-specific or time-varying in any way?

**Ethical Concerns:**

["NO or VERY MINOR ethics concerns only"]

**Final Justification:**

The proposed method appears to be effective, as demonstrated by its performance on benchmark datasets (mouse and two zebrafish in the PC forecasting setting).  the mouse and two zebrafish datasets in the PC forecasting setting). The paper is clearly written, and the empirical analysis is fairly comprehensive, though not entirely satisfying. Based on these factors, I recommend acceptance.
However, the performance gains of the proposed method are limited and, in many cases, not consistent. Notably, it often performs comparably to a simple MLP, which raises concerns about the method's overall contribution. Additional experiments and deeper analysis are necessary to better understand the strengths and limitations of the approach and to convincingly demonstrate its advantages.
Overall, I recommend borderline acceptance.

**Limitations:**

yes

**Quality:**

3

**Strengths And Weaknesses:**

Overall, I find the paper well-written and easy to follow. The proposed approach is clearly described, and the empirical analysis is fairly comprehensive.

However, while the method appears to work, I remain unconvinced about several key design choices. In particular, it is unclear why the authors chose to model population-level influences via feature-wise linear modulation. Could the authors provide more intuition and justification for this architectural decision?

Moreover, the reported performance gains of POCO over the MLP baseline appear limited in some settings. Specifically:
(a) On the two C. elegans datasets (Table 2), POCO shows only marginal or no improvement.
(b) On the two Zebrafish datasets (Table 3), the performance advantage of POCO when using PCA-reduced activity diminishes or vanishes when using single-neuron activity.

It would strengthen the paper to include a more detailed qualitative comparison—e.g., visualizing representative predictions from POCO and MLP—to better illustrate the differences in their predictions and substantiate the proposed architecture's claimed benefits.

---

> ### Author Rebuttal · Authors · 2025-07-30
>
> Thanks for your constructive feedback and thoughtful suggestions!
>
> > Prediction Visualization: To better understand the differences among the models, it would be useful to include a visual comparison of the predictions from POCO, MLP, and TexFilter—either by extending Fig. 2C or by adding a supplementary figure. This could help illustrate the qualitative distinctions that are not fully captured by the prediction metrics.
>
> Thank you for the suggestion regarding additional visualizations. We do find that these plots help clarify when POCO outperforms other models. While we are currently unable to update the figures due to conference policy, we describe the findings in text below.
>
> In many trials, we observe that the predictions of POCO, MLP, and TexFilter are highly similar. For instance, at the single-neuron level, when calcium activity fluctuates around the baseline, all models tend to produce relatively flat predictions, and none predict the small fluctuations accurately—likely because these reflect noise rather than meaningful structure. Similarly, for both PC and raw neuron predictions, all models tend to perform similarly when the context exhibits a clear trend (e.g., a slow decay following a recent calcium peak). We believe this is due to the slow dynamics of calcium signals, which all models can exploit.
>
> However, when we selectively examine trials and neurons where model predictions differ, POCO often performs better in cases where the post-context activity deviates from the baseline or preceding trend. For example, when calcium activity is flat in the context window but rises or falls afterward, both MLP and TexFilter often produce flat predictions—similar to the copy baseline—whereas POCO often more accurately captures the deviation. This suggests that POCO is more capable of leveraging population-level information to detect subtle changes in neural dynamics, rather than simply extrapolating trends from the individual neuron's past activity.
>
> > Moreover, the reported performance gains of POCO over the MLP baseline appear limited in some settings. Specifically: (a) On the two C. elegans datasets (Table 2), POCO shows only marginal or no improvement. (b) On the two Zebrafish datasets (Table 3), the performance advantage of POCO when using PCA-reduced activity diminishes or vanishes when using single-neuron activity.
>
> Thank you for pointing this out. We agree that POCO’s gains are smaller in some settings. As the five datasets come from different labs and use distinct experimental pipelines, the signal-to-noise ratio (SNR) varies across datasets. In general, simpler models may appear to perform relatively better on low-SNR recordings, as they are less prone to overfitting noise.
>
> In particular:
> (a) *C. elegans* datasets involve relatively small populations, shorter recordings, and potentially simpler dynamics, all of which can limit the benefit of population-level modeling.
> (b) For the zebrafish datasets, single-neuron activity is likely noisier than PCA-reduced signals, as PCA effectively filters out small, uncorrelated noise. Moreover, as discussed earlier, models tend to predict similarly when calcium activity fluctuates around baseline. Since neural activity is sparser in single-neuron space than in PC space—and performance is averaged across all neurons—this can numerically reduce the observed performance gap between models.
>
> While we believe these are plausible explanations, we acknowledge in the Discussion that a systematic understanding of how dataset properties influence performance remains lacking and is an important direction for future work. Notably, most recent work on neural forecasting and foundation models does not include multi-species datasets [4, 5, 6]. We see it as a strength of our study to evaluate models across diverse datasets, as this exposes both their strengths and limitations—even if it makes the overall results less “clean.”
>
> We also note that in the time-series forecasting literature, new models rarely achieve top performance across all datasets [1, 2, 3], suggesting this challenge is not unique to our setting, but reflects a broader difficulty of modeling time-series from complex systems.
>
> [1] Luo et al., *ModernTCN: A Modern Pure Convolution Structure for General Time Series Analysis*.
> [2] Yi et al., *FilterNet: Harnessing Frequency Filters for Time Series Forecasting*.
> [3] Ekambaram et al., *TSMixer: Lightweight MLP-Mixer Model for Multivariate Time Series Forecasting*.
> [4] Lueckmann et al., *ZAPBench: A Benchmark for Whole-Brain Activity Prediction in Zebrafish*.
> [5] Azabou et al., *A Unified, Scalable Framework for Neural Population Decoding*.
> [6] Zhang et al., *Towards a “Universal Translator” for Neural Dynamics at Single-Cell, Single-Spike Resolution*.
>
> > In particular, it is unclear why the authors chose to model population-level influences via feature-wise linear modulation. Could the authors provide more intuition and justification for this architectural decision?
>
> Thank you for the question. Indeed, in neuroscience, population dynamics are often modeled using recurrent architectures such as latent dynamical systems (e.g., Latent PLRNN, which we included in our benchmarks). However, we found empirically that such models are difficult to scale across heterogeneous sessions and large populations, and are also prone to overfitting.
>
> In contrast, FiLM offers a simple yet expressive mechanism to condition the forecaster by modulating its hidden activations based on the global population state. Recall that $M$ is the hidden layer size in MLP and $P = 16$ is the number of time steps we hope to predict. The final prediction $\hat{\mathbf{x}}_{t:t+P}$ can be viewed as a linear combination of $M$ temporal modes $\mathbf{v}\_1, \mathbf{v}\_2, \dots, \mathbf{v}\_M \in \mathbb{R}^{P}$, where each $\mathbf{v}\_i$ is a column of the output weight matrix $W\_{\text{out}}$. The hidden layer activations $\mathbf{h} \in \mathbb{R}^M$ control the contribution of each mode:
>
> $\hat{\mathbf{x}}\_{t:t+P} = \mathbf{b}\_{\text{out}} + \sum\_{i=1}^M \mathbf{h}\_i \mathbf{v}\_i$
>
> By using FiLM to condition $\mathbf{h}$, the population encoder effectively adjusts the contribution of each mode, tailoring each neuron’s forecast based on the broader brain state. Moreover, FiLM is computationally lightweight and scales efficiently to large neural populations since it avoids explicitly modeling pairwise interactions.
>
> > Context Length Saturation: I am also puzzled by the observation that POCO’s performance saturates rapidly with increasing context length. Could the authors provide some insight into why this might be happening?
>
> One plausible explanation is that much of the predictive information in calcium imaging traces is contained within relatively short temporal windows. From a dynamical systems perspective, the future evolution of a neural population depends on its current state. While calcium activity at a single time step may not fully reflect the underlying neural state—due to the slow decay and integration properties of calcium signals—it is likely that a relatively short history is sufficient to approximate the relevant state information for forecasting.
>
> > Learnable Latents (L0): Could the authors clarify the learnable latent variables L0? Are these latents shared across all time slices and sessions, or are they session-specific or time-varying in any way?
>
> Yes, these are shared across all time slices and sessions. In early experiments, we did try making them session-specific by adding a session embedding to L0, but it didn’t seem to result in a clear difference. We will update the manuscript to avoid confusion.

---

> > ### Comment · Reviewer_xBNB · 2025-08-03
> >
> > I thank the authors for their response, which helped clarify several points. As suggested, I encourage the authors to include a discussion on how dataset properties (e.g., species, SNR, recording length, recording size, etc.) influence model performance. Such a discussion would also aid readers in better understanding the strengths and potential limitations of the proposed method.

---

> > > ### Author Response · Authors · 2025-08-04
> > >
> > > Thank you for the thoughtful suggestions! We appreciate your encouragement to include a discussion on how dataset properties influence model performance. While we did not include an extensive discussion in the current manuscript due to space constraints, we agree that this would provide valuable context for readers—especially given the broad range of experiments presented. We plan to incorporate this discussion in the camera-ready version.

---

### Official Review · Reviewer_9twB · 2025-06-30

**Clarity:** 3
**Significance:** 4
**Originality:** 3
**Rating:** 5
**Confidence:** 5

**Summary:**

The paper proposes a neural forecasting model called POCO that includes a forecaster and an encoder based on POYO to predict future calcium imaging activity based on past calcium imaging activity. The model is tailored to spontaneous activity and is evaluated on multiple datasets recordings from different species, where it beats a wide range of baselines. The authors also show that the learned embeddings reveal brain regions and investigate factors influencing performance.

**Questions:**

These are discussion points only. I would not expect the author to actually assess any of the below points in their rebuttal.

1. The forecaster output is 1-dimensional, taking into account temporal patterns of individual neurons only. The authors point to previous work that has shown this structure to perform well. But the focus of that work was different. Would the authors expect improvements for higher dimensional forecasters?

2. While the models are evaluated using a range of performance metrics (MSE, MAE, Prediction Score), the predictor $f$ is optimized to minimize the MSE (Eq. 1). Did the authors consider different objectives? This might be particularly important for generalizing the approach to other recording types, as mentioned in the Discussion.

3. In the model, calcium imaging dynamics are not explicitly separated from underlying neuron dynamics. Was this a deliberate design decision?

**Ethical Concerns:**

["NO or VERY MINOR ethics concerns only"]

**Final Justification:**

I thank the authors for answering all of my questions. I had minor concerns to begin with and don't have any major doubts, also reading the other reviewer comments. I recommend acceptance.

**Limitations:**

The authors discussed some limitations including problems with session differences and generalization, as well as recording types beyond calcium imaging. In my view, this covers the essential points.

**Paper Formatting Concerns:**

None.

**Quality:**

4

**Strengths And Weaknesses:**

Quality:
The paper introduces a novel model architecture, which consistently outperforms competing models on spontaneous calcium activity predictions. Comparisons include a large range of baselines, evaluated on several datasets including different species and recording parameters. These comparisons look fair to me. There is also a comparison on synthetic data where the proposed model does not outperform baselines. The authors attribute this to more complex characteristics of natural data but do not explore this in detail. This would have been possible with synthetic data. The authors also find good performance on PCA reduced calcium imaging data further suggesting versatility of the proposed model. Moreover, there is an ablation study, showing the necessity of the different POCO components.

Clarity:
The manuscript is very well written, motivating and formalizing the problem well. I could not find any mistake in the equations. Some very minor points: For equation lines where two equations are separated by a comma, please increase the space after the comma to avoid confusion about the meaning. Also in Fig. 1, squares suggest the same number of elements, i.e. $M=N$ and $d=N*3$, which I believe is not the case. It would be less confusing to instead use rectangles. Moreover, exactly the same squares are used, suggesting no changes in parameters.

Significance:
The goal of this work is to develop a neural prediction foundation model that generalizes across subjects. The authors mention utility in terms of benchmark models, closed-loop optogenetic control, and whole-brain multi-animal applications. No doubt, a general neural prediction model has broad applicability in many neuroscience settings.

Originality:
The proposed model adapts POYO as a model component but goes beyond that previous model by combining POYO with a univariate forecasting model. Properties of individual neurons, as well as session properties are dealt with using embeddings from a POYO component of the model. Unlike previous approaches, the emphasis of this work is on spontaneous, task-free calcium activity. The closest previous work might be the cited reference [29] which, however, is limited to a single subject, single species analysis. The proposed POCO model does not benefit from multiple species, though (Table 4). So the main benefit is its generalization performance across subjects.

---

> ### Author Rebuttal · Authors · 2025-07-30
>
> Thanks for your positive feedback and insightful questions! As you suggested, we will increase the space after the comma in equations, change the shapes in model illustration for better clarity
>
> > The forecaster output is 1-dimensional, taking into account temporal patterns of individual neurons only. The authors point to previous work that has shown this structure to perform well. But the focus of that work was different. Would the authors expect improvements for higher dimensional forecasters?
>
> In the field of time-series forecasting, many univariate models have been shown to perform on par with or even better than multivariate models [1, 2]. However, as you rightly pointed out, these prior methods focus on different domains and were not evaluated on neural data. A key difference between neural recordings and time-series from other domains (e.g., finance, energy) is that neural channels often exhibit stronger causal interactions, potentially due to direct or indirect synaptic connections. For this reason, we expect a multivariate model to outperform simple univariate forecasters in the neural setting. This motivated our use of a population encoder to modulate otherwise simple MLP forecasters, allowing the model to capture the influence of global brain state on individual neurons.
>
> An alternative approach would be to make the MLP forecaster itself multivariate (e.g., by using the entire population as input). However, a naive implementation of this could significantly increase the number of parameters and make the model more susceptible to overfitting. A promising direction for future work could be to incorporate summary statistics (e.g., average activity of nearby neurons or neurons within the same region) as additional input to the MLP forecaster, striking a better balance between expressivity and generalization.
>
> > While the models are evaluated using a range of performance metrics (MSE, MAE, Prediction Score), the predictor is optimized to minimize the MSE (Eq. 1). Did the authors consider different objectives? This might be particularly important for generalizing the approach to other recording types, as mentioned in the Discussion.
>
> In this work, we optimized all models using mean squared error (MSE). This choice aligns with prior work in time-series forecasting and offers a stable training objective for multi-step prediction [1, 2]. However, we agree that alternative objectives—such as minimizing MAE or optimizing frequency-domain criteria—may be valuable, especially when adapting models to other types of neural recordings with different signal characteristics (e.g., spiking data or voltage imaging). Exploring alternative loss functions tailored to downstream goals or specific signal modalities (e.g., emphasizing timing accuracy or spectral fidelity) is a promising direction for future work.
>
> > In the model, calcium imaging dynamics are not explicitly separated from underlying neuron dynamics. Was this a deliberate design decision?
>
> Thank you for this insightful question. Indeed, we did not explicitly separate calcium indicator dynamics from the underlying neuronal activity. This was a deliberate design choice aimed at keeping the model architecture broadly applicable across datasets with varying sampling rates, indicators (e.g., GCaMP6s, GCaMP6f, GCaMP7f), and preprocessing pipelines. Instead of modeling deconvolved spikes or assuming a fixed calcium convolution kernel, we train directly on the observed fluorescence traces, allowing the model to implicitly learn and adapt to the dynamics present in each dataset.
>
> That said, incorporating explicit models of calcium kinetics may enhance generalization across recording modalities and possibly mitigate the challenge we found when doing multi-dataset training. This would be an interesting direction for future work, especially for bridging calcium imaging recordings with spiking datasets.
>
> [1] Yi et al. Filternet: Harnessing frequency filters for time series forecasting. [2] Ekambaram et al. Tsmixer: Lightweight mlp-mixer model for multivariate time series forecasting.

---

> > ### Comment · Reviewer_9twB · 2025-08-02
> >
> > I thank the author for their detailed discussion points, which include a wealth of additional ideas. I don't have further questions.

---

### Decision · Program_Chairs · 2025-09-17

**Decision:**

Accept (poster)

**Comment:**

The paper introduces a new model for forecasting calcium activity, evaluated on four custom datasets and the ZAPBench benchmark. On the custom datasets, the model outperforms several baselines, while on ZAPBench, its performance is comparable to existing methods. The authors release their code and two of the four datasets, providing a valuable benchmark setting for neural forecasting research.

Reviewers are broadly positive about the paper’s contribution. Reviewer 9twB highlighted cross-subject generalization and asked mostly clarification questions. Reviewer xBNB was more critical, questioning architectural choices and the modest improvements over baselines, which the authors acknowledged. Reviewer KEzN raised issues about baseline tuning and experiments, but was convinced by the authors’ additional results, while also noting limited novelty relative to prior work (e.g., POYO, NDT). Reviewer pkKL emphasized concerns about clarity, particularly in how the contribution is framed in the abstract, introduction, and discussion.

Post-review, the paper received borderline acceptance recommendations. Remaining concerns include limited and inconsistent performance gains, with a simple MLP baseline not consistently surpassed. Reviewers requested further experiments and analysis; while some were provided, they were only partially satisfactory.

The central question for acceptance is whether the problem formulation and benchmarking setup justify publication. Reviewers noted that forecasting in low-frequency calcium imaging is understudied, and the paper’s main value lies in applying established methods with a thorough comparative evaluation in this domain.

Requested changes include, but are not limited to:

- "discussion on how dataset properties (e.g., species, SNR, recording length, recording size, etc.) influence model performance" (xBNB)
- include additional experimental results discussed with Reviewer KEzN; tables critical to the discussion should be added to the appendix, and referenced/interpreted in the paper (it is not acceptable to "bury" them in the appendix given the importance to the questions about baseline tuning etc)